# Integrating genomic information and productivity and climate-adaptability traits into a regional white spruce breeding program

Eduardo P. Cappa[1,2]*, Jennifer G. Klutsch[3¤a], Jaime Sebastian-Azcona[3¤b], Blaise Ratcliffe[4], Xiaojing Wei[3], Letitia Da Ros[5], Yang Liu[4], Charles Chen[6], Andy Benowicz[7], Shane Sadoway[8], Shawn D. Mansfield[5], Nadir Erbilgin[3], Barb R. Thomas[3], Yousry A. El-Kassaby[4]*

**1** Instituto de Recursos Biológicos, Centro de Investigación en Recursos Naturales, Instituto Nacional de Tecnología Agropecuaria (INTA), Hurlingham, Buenos Aires, Argentina, **2** Consejo Nacional de Investigaciones Científicas y Técnicas (CONICET), Buenos Aires, Argentina, **3** Department of Renewable Resources, University of Alberta, Edmonton, Alberta, Canada, **4** Department of Forest and Conservation Sciences, Faculty of Forestry, University of British Columbia, Vancouver, British Columbia, Canada, **5** Department of Wood Science, Faculty of Forestry, University of British Columbia, Vancouver, British Columbia, Canada, **6** Department of Biochemistry and Molecular Biology, Oklahoma State University, Stillwater, Oklahoma, United States of America, **7** Forest Stewardship and Trade Branch, Alberta Agriculture and Forestry, Edmonton, Alberta, Canada, **8** Blue Ridge Lumber Inc., West Fraser Mills Ltd, Blue Ridge, Alberta, Canada

¤a Current address: Department of Forestry, New Mexico Highlands University, Las Vegas, New Mexico, United States of America
¤b Current address: Irrigation and Crop Ecophysiology Group, Instituto de Recursos Naturales y Agrobiología de Sevilla, Sevilla, Spain
* cappa.eduardo@inta.gob.ar (EPC); y.el-kassaby@ubc.ca (YAE)

**Data Availability Statement:** Genotyping-by-sequencing (GBS) raw reads used in this study

## Abstract

Tree improvement programs often focus on improving productivity-related traits; however, under present climate change scenarios, climate change-related (adaptive) traits should also be incorporated into such programs. Therefore, quantifying the genetic variation and correlations among productivity and adaptability traits, and the importance of genotype by environment interactions, including defense compounds involved in biotic and abiotic resistance, is essential for selecting parents for the production of resilient and sustainable forests. Here, we estimated quantitative genetic parameters for 15 growth, wood quality, drought resilience, and monoterpene traits for *Picea glauca* (Moench) Voss (white spruce). We sampled 1,540 trees from three open-pollinated progeny trials, genotyped with 467,224 SNP markers using genotyping-by-sequencing (GBS). We used the pedigree and SNP information to calculate, respectively, the average numerator and genomic relationship matrices, and univariate and multivariate individual-tree models to obtain estimates of (co) variance components. With few site-specific exceptions, all traits examined were under genetic control. Overall, higher heritability estimates were derived from the genomic- than their counterpart pedigree-based relationship matrix. Selection for height, generally, improved diameter and water use efficiency, but decreased wood density, microfibril angle, and drought resistance. Genome-based correlations between traits reaffirmed the pedigree-

have been deposited in NCBI SRA BioProject - PRJNA748443 (https://www.ncbi.nlm.nih.gov/bioproject/PRJNA748443). Information of the white spruce trials including pedigree and adjusted and standardized phenotypic data are available in the GitHub repository: https://github.com/RESFOR/quantitative_genetics_R/blob/e067422f5e56ec7bb98e4265e60e875603bf51b5/White_Spruce_Phenotype_Pedigree_PLoS2022.TXT".

**Funding:** This work was funded by Genome Canada (https://www.genomecanada.ca/) RES-FOR ID 10207, grants 16R75036 to YAE, RES0034654 to NE, and RES0031330 to BRT; Genome Alberta (https://genomealberta.ca/) RES-FOR ID: LRF, grants RES0034664 to NE, 16R10106 to SDM, and RES0034657 to BRT; University of Alberta / Faculty ALES / Dept RR (https://www.ualberta.ca/index.html) grant RES0034569 to BRT; Alberta Innovates – BioSolutions (https://albertainnovates.ca/) grants RES0035327 to NE, 16R75221 to SDM, and RES0028979 to BRT; Genome BC (https://www.genomebc.ca/) grants 16R75421 to YAE and 16R75546 to SDM; Forest Resource Improvement Association of Alberta (FRIAA, https://friaa.ab.ca/) grants RES0037021 and RES0036845 to BRT; National Science Foundation (NSF, tps://www.nsf.gov/) grants MRI-1531128, ACI-1548562, and ACI-1445606 to CC; The Extreme Science and Engineering Discovery (XSEDE, https://xras.xsede.org/public/requests/29304-XSEDE-MCB180177) grant MCB180177 to CC. The funders had no role in study design, data collection and analysis, decision to publish, or preparation of the manuscript.

**Competing interests:** The authors have declared that no competing interests exist.

based correlations for most trait pairs. High and positive genetic correlations between sites were observed (average 0.68), except for those pairs involving the highest elevation, warmer, and moister site, specifically for growth and microfibril angle. These results illustrate the advantage of using genomic information jointly with productivity and adaptability traits, and defense compounds to enhance tree breeding selection for changing climate.

## Introduction

White spruce (*Picea glauca* (Moench) Voss) is one of the most widely distributed North American conifer species and commercially, one of the most important tree species in the Province of Alberta (Canada) [1]. To date, most forest tree' quantitative genetic studies and tree improvement programs are primarily focused on economically important productivity traits (productivity-related traits), such as growth and wood quality (e.g., [2–4]). However, the ongoing rapid climate change resulting in higher frequency and severity of drought events has begun to change the focus of selection. In addition to directly affecting tree productivity, drought can have a profound effect on tree susceptibility to pests and pathogens [5]. Therefore, climate change-related (adaptive) traits including plasticity and adaptation to drought, forest pest and pathogens resistance should be incorporated into existing tree breeding programs [6, 7]. Aligning with this recommendation, and reviewing 260 global tree pest and disease resistance initiatives, Yanchuk and Allard [8] reported very few tree improvement programs that operationally succeeded in deploying resistant material. Moreover, for a better understanding of the interplay between productivity- and adaptability-related traits, breeders need to study which secondary compounds are associated with these traits and understand their inherent variation.

In the context of global climate change, knowledge of traits' variance components and their genetic parameters such as heritability and correlations between productivity-, adaptability-related traits, and chemical compounds related to defense and drought stress, are vital for the development of effective tree breeding programs. Moreover, either the simultaneous maximization/optimization of potential genetic gain for multiple traits, or understanding the genetic × environment (G×E) interaction from multiple site analyses, are essential to increasing tree resilience toward environmental perturbations, and for ensuring the sustainable long-term genetic progress of a breeding program [9]. Several studies have reported pedigree-based (see below) genetic parameters for productivity-related traits, such as growth and wood quality in white spruce [3, 4, 10–16] as well as pest resistance traits [17–20]. However, few studies have focused on drought resilience [21] and defense chemical traits [22]. Therefore, the genetic control, cross-site stability (i.e., G×E), and correlation of most of these adaptability-related traits remain to be understood.

To obtain precise genetic parameter estimates (or function of them), accurate information of individuals' genealogy is required [23]. The individual-tree mixed model utilizes individuals' contemporary pedigree information to estimate the additive genetic variance using Henderson's average numerator relationship matrix (*A*-matrix) [24]. However, the *A*-matrix estimates ignore all historical relationships beyond that of the contemporary pedigree as all relationships are based on identity-by-descent rather than actual relationships [25]. Thus, the accuracy of genetic parameter and predicted breeding value rankings are compromised [26]. On the other hand, the use of genomic information through molecular markers to infer the realized genomic relationship matrix (*G*-matrix; [27]) offers an efficient alternative to constructing the

additive relationship matrix, and effectively estimating individuals' realized genetic related-ness. Recently, studies in forest trees have tested the value of molecular markers for estimating genetic parameters using the *G*-matrix [2–4, 10, 28, 29]. However, these studies only focused on growth and/or wood quality traits and limited work examined drought resilience, and/or pest resistance via chemical defense traits [20, 30].

As part of a large-scale tree genomic study [31] we genotyped 1,540 white spruce trees with 467,224 SNPs and phenotyped them for various productivity-, adaptability-related traits, and defense monoterpenes. These trees represent a subset of open-pollinated progeny being tested and grown on multiple genetic test sites throughout the Province of Alberta [31]. The available genotypic and phenotypic information for these trees offered a unique opportunity to evaluate the genetic control and relationships of the assessed traits, and the extent of G×E interactions. Here, we studied 15 growth, wood quality, drought resilience, and defense and drought stress chemical traits (monoterpenes), and estimated their quantitative genetic parameters (includ-ing heritability and genetic correlations) within and across-sites. Estimates were obtained and compared using both pedigree- and genomic-based relationship matrices. The results of this study are expected to provide critical information needed for the identification and selection of genetic material for their inclusion in new production populations (seed orchards). New second generation orchards will replace the aging first generation orchards which currently supply 65% of all white spruce reforestation stock in Alberta (Andy Benowicz, personal com-munication). There is an urgent need to change the orchard production profiles from the cur-rent ones focused on improved growth only, to the ones focused on improved climate resiliency.

## Materials and methods

### Genetic material and trial description

Three open-pollinated (OP) progeny trials (Calling Lake: CALL, Carson Lake: CARS, and Red Earth: REDE) of the Alberta Agriculture and Forestry white spruce Region D1 breeding pro-gram [32] were used in this study (Table 1 and Fig 1). These trials were planted in a random-ized complete block design with six replicates and 5- or 6-tree row plots at 2.5× 2.5 m spacing (Table 1). The entire population being tested in the three progeny trials consisted of 150 fami-lies from 10 provenances. Based on age-30 tree height, a sub-sample of 80 families were selected representing low-, average- and high-class heights, each with approximately eight individual progeny per family for CALL and REDE, and four progeny for CARS (*n* = 1,483). An additional 142 potential forward selected trees, previously identified in the three progeny trials and based on height breeding values, were also included for sequencing. From these 142 forward selected trees, 34 trees were from an additional 19 families, resulting in a total of 1,625 trees from 99 families.

### Traits evaluated

Diameter at breast height (1.3 m; DBH) and tree height (HT) were measured at age-30 and represent the growth productivity traits measured. Wood density (WD) was measured using a 5 mm bark to pith increment cores taken close to breast height on the north facing side of each tree. Cores were transported in straws to the lab, soxhlet extracted overnight with hot acetone, precision cut to 1.68 mm thickness with a twin blade pneumatic saw, and allowed to acclimate to 7% moisture before density analysis. All samples were then scanned from pith to bark by X-ray densitometry (Quintek Measurement Systems, TN) at a resolution of 0.0254 mm. We report data as relative density on an oven-dry weight basis. Finally, average WD was calculated as the weighted WD of the individual tree rings weighted by their annual basal area increment

**Table 1. Trial location, sites and climate characteristics, date of planting, experimental design data, and number of original trees selected in each of the three open-pollinated white spruce trials.**

| Trial[a] | CALL | CARS | REDE |
|---|---|---|---|
| **Location** | Calling Lake | Carson Lake | Red Earth |
| **Latitude (˚N)** | 55˚16' | 54˚34' | 56˚34' |
| **Longitude (˚W)** | 113˚ 09' | 115˚34' | 115˚19' |
| **Elevation (m)** | 640 | 1006 | 518 |
| **Soil texture** | Clay loam | Clay loam | Clay loam |
| **MAT (˚C)** | 1.6 | 2.9 | 1.3 |
| **MWMT (˚C)** | 16.3 | 15.0 | 16.6 |
| **MAP (mm)** | 467 | 535 | 442 |
| **MSP (mm)** | 327 | 371 | 300 |
| **CMI (mm)** | 2.2 | 13.1 | -0.5 |
| **Planting date** | May-1986 | May-1987 | May-1987 |
| **Number of replicates** | 6 | 6 | 6 |
| **Number of tree per plot** | 5 | 6 | 6 |
| **Number of rows** | 52 | 60 | 61 |
| **Number of columns** | 120 | 102 | 96 |
| **Initial number of trees** | 4380 | 5292 | 5400 |
| **Survival at 30 years (%)** | 90 | 77 | 94 |
| **Number of trees selected** | 647 | 314 | 603 |

[a] MAT mean annual temperature; MWMT mean warmest month temperature; MAP mean annual precipitation; MSP mean annual summer (May to Sept.) precipitation; CMI Hogg's climate moisture index. The climate variables represent the average over the study period 1986–2019, and are based on ClimateBC v7.00 [33].

(*BAI*) to better represent the density of the whole tree. Given that juvenile rings have less reliable measurements we discarded tree rings prior to 1995. Microfibril angle (MFA) was determined by X-ray diffraction by determining the 002 diffraction arc (T-values) using a Bruker D8 Discover X-ray diffraction unit equipped with an area array detector (GADDS) on the radial face of the individual growth rings, as previously detailed by Ukrainetz et al. [36].

Two dendrochronological indices were calculated from tree ring information: drought resistance (Resistance) and mean drought sensitivity (Sensitivity). Resistance represents the ability of a tree to maintain growth during a specific drought episode, in this case it occurred in 2015, and was calculated using the following equation [37]: $Resistance = BAI_{drought}/BAI_{pre-drought}$, where $BAI_{drought}$ is the average *BAI* of the drought event (2015) and $BAI_{pre-drought}$ is the average *BAI* of the four years before the drought event (2011–2014) (see S1 Fig). Resistance describes how much the incremental growth is reduced during a drought event. As such, a Resistance value close to 1 represents a tree unaffected by the drought, while smaller values represent less resistant trees. Sensitivity is a classic dendrochronological index commonly used to estimate the responsiveness of trees to climate [38], and was calculated as:

$$Mean\ Sensitivity = \frac{1}{n-1} \sum_{t=1}^{t=n-1} \left| \frac{2(BAI_{t+1} - BAI_t)}{BAI_{t+1} + BAI_t} \right|$$

where, $BAI_t$ is the *BAI* measured at year *t* and *n* is the total number of years measured. Trees with high climatic sensitivity are able to grow particularly well under good environmental conditions but are more severely affected by drought events.

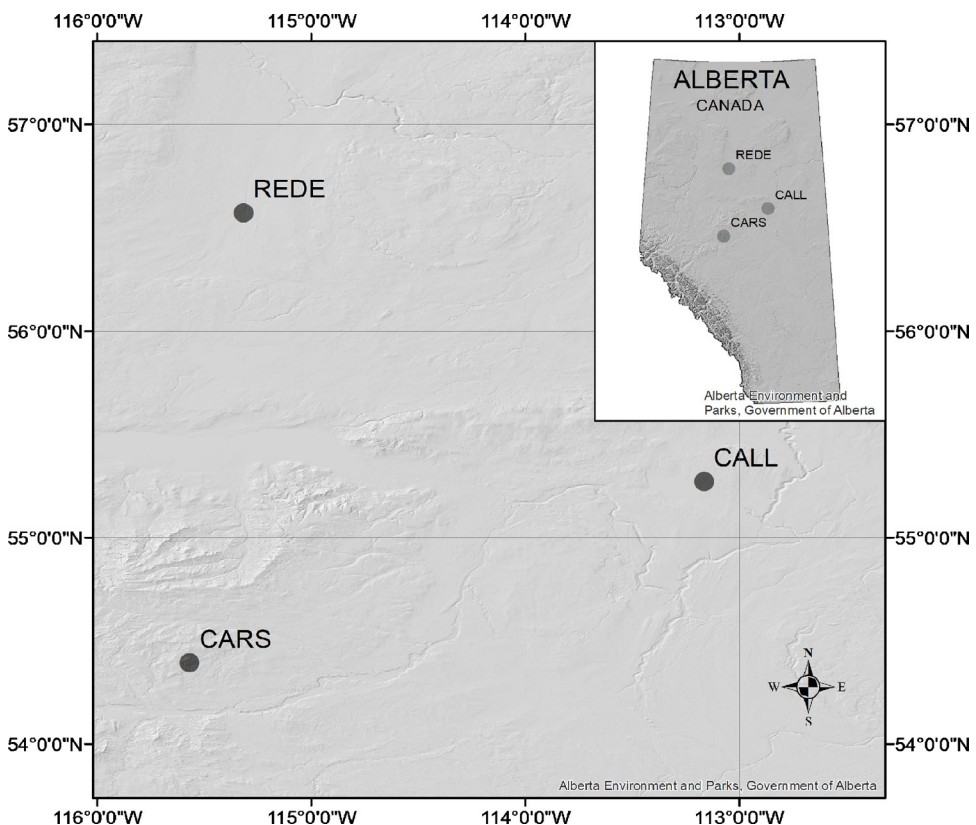

**Fig 1. Location of the three white spruce (grey circles) test sites in Alberta, Canada.** Abbreviations used for the test sites are described in the Table 1. This figure was created in ArcMap [34] using Government of Alberta data [35].

The two residual outside pieces of the cores (slabs), retained during pneumatic processing of the density specimens along the increment cores radial direction of the cross section, were used to capture the variation in the stable carbon isotope ratio ($\delta^{13}C$) across all years measured on each tree. The slabs were dried and ground using a Qiagen TissueLyser II (Qiagen Inc., Hilden, Germany). During grinding, each sample was placed into an individual stainless-steel jar with a 2 cm stainless-steel ball. The ground samples were then analyzed for $\delta^{13}C$ at Alberta InnoTech Stable Isotope Laboratory, Victoria, Canada. The analysis was performed using an established method on a MAT 253 mass spectrometer with Conflo IV interface (Thermo Fisher Scientific, Waltham, MA, USA.) and a Fisons NA1500 EA (Fisons Instruments, Milano, Italy). In brief, approximately 1 mg of solid sample was weighed into tin capsules then placed into a combustion reactor that produces $CO_2$, which was then analyzed by mass spectrometry for isotopic estimates. Multiple in-house standards, calibrated relative to international standards, were run to allow the results to be normalized and reported vs. Vienna Pee Dee Belemnite. $\delta^{13}C$ values were used as a measure of intrinsic long-term water use efficiency (WUE).

The defense compounds identified and quantified were mainly monoterpenes assessed from needles collected from south facing branches near the crown of the trees during May—June (2017), and from the 99 families selected and studied across all three test sites ($n = 1,602$) (see S1 Text "Chemical analysis" for details). Briefly, needle samples were kept at -40˚C and ground to a powder for extraction. Hexane-extractable compounds were identified and quantified with a gas chromatography-flame ionization detector using methods modified from [39]. We identified 12 hexane-extractable compounds and used seven monoterpenes ($\alpha$-pinene, $\beta$-pinene, camphene, myrcene, limonene, terpinolene, and camphor), including the

**Table 2. Phenotypic mean for the 15 traits assessed in the white spruce population.**

| Trait | Unit | *n* | Mean | SD | CV | Min. | Max. |
|---|---|---|---|---|---|---|---|
| HT | cm | 1,516 | 947.32 | 1.72 | 0.18 | 200 | 1350 |
| DBH | cm | 1,516 | 14.94 | 3.32 | 0.22 | 1.6 | 26 |
| WD | kg.m$^{-3}$ | 1,448 | 377.32 | 28.94 | 0.08 | 304.07 | 497.64 |
| MFA | ° | 1,510 | 21.18 | 3.93 | 0.19 | 17.15 | 56.79 |
| Resistance | - | 1,435 | 0.57 | 0.14 | 0.25 | 0.23 | 1.33 |
| Sensitivity | - | 1,445 | 0.23 | 0.07 | 0.30 | 0.03 | 0.45 |
| δ$^{13}$C | - | 1,509 | -25.9 | 0.68 | -0.03 | -28.14 | -23.55 |
| α-Pinene | ng mg$^{-1}$ | 1,418 | 169.67 | 151.33 | 0.89 | 13.99 | 1502.32 |
| β-Pinene | ng mg$^{-1}$ | 932 | 30.49 | 20.02 | 0.66 | 8.18 | 215.51 |
| Camphene | ng mg$^{-1}$ | 1,362 | 367.39 | 356.03 | 0.97 | 10.43 | 2585.76 |
| Camphor | ng mg$^{-1}$ | 1,183 | 758.19 | 677.58 | 0.89 | 17.79 | 5769.53 |
| Myrcene | ng mg$^{-1}$ | 1,472 | 358.54 | 377.38 | 1.05 | 13.79 | 5644.61 |
| Limonene | ng mg$^{-1}$ | 1,472 | 429.13 | 425.42 | 0.99 | 10.9 | 3590.68 |
| Terpinolene | ng mg$^{-1}$ | 906 | 39.36 | 22.76 | 0.58 | 8.17 | 169.09 |
| Total monoterpenes | ng mg$^{-1}$ | 1,495 | 2934.11 | 2425.41 | 0.83 | 13.1 | 18719.14 |

Number of trees for which trait values were used in the quantitative parameters analyses (*n*), and statistics: mean, standard deviation (SD), phenotypic coefficient of variation (CV), minimum (Min.), and maximum (Max.) values observed. Abbreviations used for the traits are described in the text. Monoterpene concentrations are reported on a dry weight basis.

sum of all hexane-extractable compound concentrations (total monoterpenes), in the characterization of genetic parameters. Many of these compounds can be anti-feedants for *Choristoneura fumiferana* (eastern spruce budworm; [40–42]). The remaining chemical compounds did not fit model assumptions and were not included in the analyses.

Logarithmic transformations were applied to MFA and all monoterpene compounds to improve data normality (see S2 Fig). Additionally, prior to the multivariate analyses, all the phenotypic data were spatially adjusted [43] using the design effects. Design adjusted phenotypic data were obtained for each tree for each trait and site by subtracting the estimated replication effects from the original phenotype. Finally, data of all traits were standardized (mean = zero and variance = 1). The list of traits, number of trees for each trait, and summary statistics for all the phenotypic traits in their original scale (i.e., without design adjustment) are presented in Table 2.

## Genotyping-by-sequencing

Following Chen´s et al. [44] genotyping-by-sequencing (GBS) protocol, the DNA from each needle sample was prepared with EcoT22-I (ATGCA) restriction enzyme digestion. Sequencing reads of 1,625 trees were aligned to the most up-to-date white spruce assembly (WS77111-v2, [45]) using BWA [46] and TASSEL-GBS [47]. Of the total 30 million SNP read tags constructed, ~ 26 million tags (87.5%) were aligned to the genome assembly and 4.5 million SNPs were determined with an individual site depth at 4x coverage. A set of 1,599 trees and 467,224 (467K) biallelic SNPs were obtained based on filtering the SNP data set for a maximum missing data proportion of 30%, a minor allele count of one, and maximum site read depth < = 70. Missing data were imputed using the mean observed allele at each locus.

## Pedigree correction

Using the filtered SNP subset, we validated and corrected the pedigree of the OP families based on the comparison of the expected versus observed additive genetic relationships using a

custom R-script. Samples' pairwise additive relationship coefficients of the $G$-matrix (see below) were examined for large deviations from their expected values (e.g. 0.25 for half-sib) and corrected parentage was assigned or reassigned manually.

We removed 59 sampled trees for parent conflicts. Of the final set of 1,540 trees, 202 trees' pedigree records were modified or corrected. These changes mostly stemmed from the identification of 5 phantom mothers and 100 pollen donors (fathers), which increased the number of identified parents for the 1,540 white spruce trees from 99 (original pedigree) to 204 (corrected pedigree). The number of genotyped trees per mother had a range of 1–20, and from 1 to 8 per site.

## Quantitative genetics analyses

Our single-trait single-site analysis used a univariate individual-tree mixed model as following:

$$\boldsymbol{y} = \boldsymbol{X\beta} + \boldsymbol{Z_d d} + \boldsymbol{Z_a a} + \boldsymbol{e} \tag{1}$$

where, $\boldsymbol{y}$ is the vector of phenotypic data; $\boldsymbol{\beta}$ is the vector of fixed effects genetic groups formed according to provenances; $\boldsymbol{d}$ is the vector of random design effects, including replications, however, given that in general just one RES-FOR trees was sampled from each 4-tree row plot, the plot effects were not fitted; $\boldsymbol{a}$ is the vector of random genetic effects following a normal distribution with zero mean and (co)variance matrix $\boldsymbol{A}\sigma_a^2$, where $\boldsymbol{A}$ is the average numerator relationship matrix and $\sigma_a^2$ is the additive genetic variance; and $\boldsymbol{e}$ is the vector of the random residual effect following also a normal distribution with zero mean and (co)variance matrix $\boldsymbol{I}\sigma_e^2$, where $\boldsymbol{I}$ is the identity matrix and $\sigma_e^2$ is the residual error variance. $\boldsymbol{X}$, $\boldsymbol{Z_d}$, and $\boldsymbol{Z_a}$, are incidence matrices relating fixed and random effects to measurements in vector $\boldsymbol{y}$.

Genetic correlations between different traits measured from the same individual, and genetic correlations between sites, considering measurements from different sites as different traits, were estimated based on the following multiple-trait individual-tree mixed model:

$$\begin{bmatrix} \boldsymbol{y}_i \\ \vdots \\ \boldsymbol{y}_j \end{bmatrix} = \begin{bmatrix} \boldsymbol{X}_i & \cdots & \boldsymbol{0} \\ \vdots & \ddots & \vdots \\ \boldsymbol{0} & \cdots & \boldsymbol{X}_j \end{bmatrix} \begin{bmatrix} \boldsymbol{\beta}_i \\ \vdots \\ \boldsymbol{\beta}_j \end{bmatrix} + \begin{bmatrix} \boldsymbol{Z}_{ai} & \cdots & \boldsymbol{0} \\ \vdots & \ddots & \vdots \\ \boldsymbol{0} & \cdots & \boldsymbol{Z}_{aj} \end{bmatrix} \begin{bmatrix} \boldsymbol{a}_i \\ \vdots \\ \boldsymbol{a}_j \end{bmatrix} + \begin{bmatrix} \boldsymbol{e}_i \\ \vdots \\ \boldsymbol{e}_j \end{bmatrix} \tag{2}$$

where, $[\boldsymbol{y}_i' | \cdots | \boldsymbol{y}_j']$ included the individual-tree spatially adjusted phenotypes for all traits and sites; the genetic groups effects for each trait or site are included in $[\boldsymbol{\beta}_i' | \cdots | \boldsymbol{\beta}_j']$; the genetic effects (breeding values) of all individuals for all the traits or sites are included in $[\boldsymbol{a}_i' | \cdots | \boldsymbol{a}_j']$, and $[\boldsymbol{e}_i' | \cdots | \boldsymbol{e}_j']$ is the residual vector. The incidence matrices $\boldsymbol{X}_i \oplus \cdots \oplus \boldsymbol{X}_j$, and $\boldsymbol{Z}_{a_i} \oplus \cdots \oplus \boldsymbol{Z}_{a_j}$ related observations in $[\boldsymbol{y}_i' | \cdots | \boldsymbol{y}_j']$ to elements of $[\boldsymbol{\beta}_i' | \cdots | \boldsymbol{\beta}_j']$ and $[\boldsymbol{a}_i' | \cdots | \boldsymbol{a}_j']$, respectively. The symbols $\oplus$ and ' indicates the direct sum of matrices and transpose operation, respectively. Finally, the expected value and variance-covariance matrix of the genetic effects in model (2) are respectively equal to:

$$E \begin{bmatrix} \boldsymbol{a}_i \\ \vdots \\ \boldsymbol{a}_j \end{bmatrix} = \begin{bmatrix} \boldsymbol{0} \\ \vdots \\ \boldsymbol{0} \end{bmatrix}, \quad Var \begin{bmatrix} \boldsymbol{a}_i \\ \vdots \\ \boldsymbol{a}_j \end{bmatrix} = \begin{bmatrix} \sigma_{a_{ii}}^2 \boldsymbol{A} & \cdots & \sigma_{a_{ij}} \boldsymbol{A} \\ \vdots & \ddots & \vdots \\ \sigma_{a_{ji}} \boldsymbol{A} & \cdots & \sigma_{a_{jj}}^2 \boldsymbol{A} \end{bmatrix} = \begin{bmatrix} \sigma_{a_{ii}}^2 & \cdots & \sigma_{a_{ij}} \\ \vdots & \ddots & \vdots \\ \sigma_{a_{ji}} & \cdots & \sigma_{a_{jj}}^2 \end{bmatrix} \otimes \boldsymbol{A}$$

where, $\sigma_{a_{ii}}^2$ and $\sigma_{a_{jj}}^2$ are the genetic variances for the traits or sites $i$ and $j$ respectively, $\sigma_{a_{ij}}$ is the genetic covariance between traits or sites $i$ and $j$, and $\boldsymbol{A}$ is defined above for the single-trait

single-site model. The symbol $\otimes$ indicates the Kronecker products of matrices. The expected value and variance-covariance matrix of $e$ are equal to:

$$E\begin{bmatrix} e_i \\ \vdots \\ e_j \end{bmatrix} = \begin{bmatrix} 0 \\ \vdots \\ 0 \end{bmatrix}, \quad Var\begin{bmatrix} e_i \\ \vdots \\ e_j \end{bmatrix} = \begin{bmatrix} \sigma^2_{e_{ii}}I & \cdots & \sigma_{ae_{ij}}I \\ \vdots & \ddots & \vdots \\ \sigma_{e_{ji}}I & \cdots & \sigma^2_{e_{jj}}I \end{bmatrix} = \begin{bmatrix} \sigma^2_{e_{ii}} & \cdots & \sigma_{ae_{ij}} \\ \vdots & \ddots & \vdots \\ \sigma_{e_{ji}} & \cdots & \sigma^2_{e_{jj}} \end{bmatrix} \otimes I$$

The residual variances for traits or sites $i$ and $j$ were $\sigma^2_{e_i}$, and $\sigma^2_{e_j}$, respectively, $\sigma_{e_{ij}}$ is the residual covariance between traits $i$ and $j$, and $I$ is the identity matrix. Given that the sites were assessed separately, the residual covariances across-sites were assumed to be zero.

In the genomic-based approach, the pedigree-based relationship matrices $A$ ($A$-matrix) for genetic effects, of the previous mixed models (1) and (2), were substituted by the corresponding genomic relationship matrix ($G$-matrix) based on 467K SNPs.

$$G = \frac{WW'}{2\sum p_i(1 - p_i)}$$

where, $W$ is the $n \times m$ ($n$ = number of individuals, $m$ = number of SNPs) rescaled genotype matrix following $M$—$P$, where $M$ is the genotype matrix containing genotypes coded as 0, 1, and 2 according to the number of alternative alleles, and $P$ is a vector of twice the allelic frequency, $p_i$.

Estimates of pedigree- and genomic-based variances for the genetic effects ($\hat{\sigma}^2_a$,) and residual errors ($\hat{\sigma}^2_e$), were re-parameterized to individual-trait narrow-sense heritability ($\hat{h}^2$) and genetic correlations ($\hat{r}_a$) between traits, or sites $i$ and $j$, as follows:

$$\hat{h}^2 = \frac{\hat{\sigma}^2_a}{\hat{\sigma}^2_a + \hat{\sigma}^2_e}; \quad \hat{r}_a = \frac{\hat{\sigma}_{a_{i,j}}}{\sqrt{\hat{\sigma}^2_{a_{i,i}} \times \hat{\sigma}^2_{a_{j,j}}}}$$

Visualization of genetic correlations between traits was done using the corrplot function in R-package corrplot [48]. Correlations between traits or sites were considered strong if $\hat{r}_a \geq 0.70$, moderate if $0.70 > \hat{r}_a > 0.40$, and low or weak when $\hat{r}_a \leq 0.4$.

Univariate model (1) and multivariate model (2) were fitted in R (www.r-project.org) with the function remlf90 from the package 'breedR' [49], using the Expectation-Maximization (EM) algorithm followed by one iteration with the Average Information (AI) algorithm to compute the approximated standard errors of the variance components [50]. The remlf90 function in the R-package 'breedR' is based in the REMLF90 (for the EM algorithm) and AIR-EMLF90 (for the AI algorithm) of the BLUPF90 family [51]. The program preGSf90, also from the BLUPF90 family [51], was used to create the inverse of the $G$-matrices calculated with the 467K SNPs markers, and then used to fit models (1) and (2) with the 'breedR' package.

## Results

### Pedigree- and genomic-based relationship estimations

To study the expected (pedigree) and realized (genomic) relationship structures in the genotyped population, individual pairwise relatedness was estimated using either genome-wide marker data or pedigree (after correction) to determine the proportion of self-relationship (1.00 relatedness), full-sibs (0.50), half-sibs (0.25), and unrelated (0.00) individuals. For the 1,540 genotyped trees, we determined a total of 2,371,600 pairwise relationships. After pedigree correction, the value distribution showed that 98.81% (2,343,490) of which involved

**Table 3. Statistics of pairwise relatedness coefficients.** Statistics of pairwise relatedness coefficients for self-relationship coefficients, full-sibs, and half-sibs and unrelated genotyped trees, for both the pedigree (after pedigree correction *A*-matrix) and genomic information from all available SNPs (467K) (*G*-matrix).

| | Self-relationships | | Full-sib | | Half sibs | | Unrelated | |
|---|---|---|---|---|---|---|---|---|
| | *A*-matrix | *G*-matrix | *A*-matrix | *G*-matrix | *A*-matrix | *G*-matrix | *A*-matrix | *G*-matrix |
| *n* | 1540 | 1540 | 360 | 360 | 26210 | 26210 | 2343490 | 2343490 |
| Mean | 1.000 | 1.120 | 0.500 | 0.377 | 0.250 | 0.176 | 0.000 | -0.002 |
| Minimum | 1.000 | 0.628 | 0.500 | 0.205 | 0.250 | 0.045 | 0.000 | -0.027 |
| Maximum | 1.000 | 1.281 | 0.500 | 0.517 | 0.250 | 0.405 | 0.000 | 0.283 |
| SD | 0.000 | 0.080 | 0.000 | 0.059 | 0.000 | 0.043 | 0.000 | 0.005 |
| CV | 0.000 | 0.071 | 0.000 | 0.156 | 0.000 | 0.243 | 0.000 | -2.547 |

Number of relationships (*n*), mean (**Mean**), minimum value (**Minimum**), maximum value (**Maximum**), standard deviation (**SD**), and coefficient of variation (**CV**).

estimates for unrelated individuals (according to the pedigree), while half-sibs represented 1.11% (26,210) and full-sibs 0.02% (360) (Table 3). A comparison of the pedigree expected and genomic realized relationship matrices is also depicted using the distribution of the number of pairwise additive relationships (S3 Fig). A good pedigree control in the production of the unrelated, half-sib and full-sib families is shown, although SNP marker data, by capturing the realized genetic relationships, provided considerably more refined estimates of the continuous distribution of true relatedness in the genotyped population.

## Heritability estimates

Overall, narrow-sense heritability estimates based on genomic relationship matrices were generally (35 out of 42 site-trait combinations) higher than those based on the pedigree relationship matrices (average of 0.54 and 0.43 across traits and sites, respectively; Fig 2). However, standard errors for heritability estimates were found to be lower for the pedigree- (0.16 averaged across traits and sites) versus genomic-based (0.19) models (Table 4).

Across test sites and relationship matrices, heritability estimates for growth traits (HT and DBH) ranged from low to high with an average estimate of 0.73 (range: 0.06–0.97). Wood quality traits (WD and MFA) showed low to moderate narrow-sense heritability estimates, averaging 0.34 (range: 0.05–0.78). Among the test sites, CARS showed significantly lower heritability estimates for DBH and WD, and the lowest MFA heritability estimate was found at the REDE. Both dendrochronological drought indices, Resistance and Sensitivity, showed moderate to high heritability estimates for CALL and REDE with values ranging from 0.25 to 0.80 (average 0.49). However, these values were near zero for CARS, i.e., with no heritable variation (additive genetic variation). For the trait $\delta^{13}C$, moderate to high heritability estimates were found with values ranging from 0.57 to 0.98 (average 0.81). Heritability estimates for monoterpene compounds, however, showed a lack of consistency, with values ranging from 0.00 to 0.96 (averaged of 0.45). Total monoterpenes showed slightly lower heritability estimates than the individual monoterpenes, ranging from 0.08 to 0.64 (average 0.39). Again, CARS showed lower heritability estimates than the other white spruce test sites for total monoterpenes (see Table 4 for details).

## Traits genetic correlations

Overall, genomic-based relationship genotypic correlation estimates are equivalent to those from the classical pedigree-based relationship with a similar average (of 0.23) across the 105 trait-pair combinations; and varied from -0.81 to 0.99 and -0.79 to 1.00, for pedigree- and genomic-based genetic correlation estimates, respectively (Fig 3 and S1 Table). However,

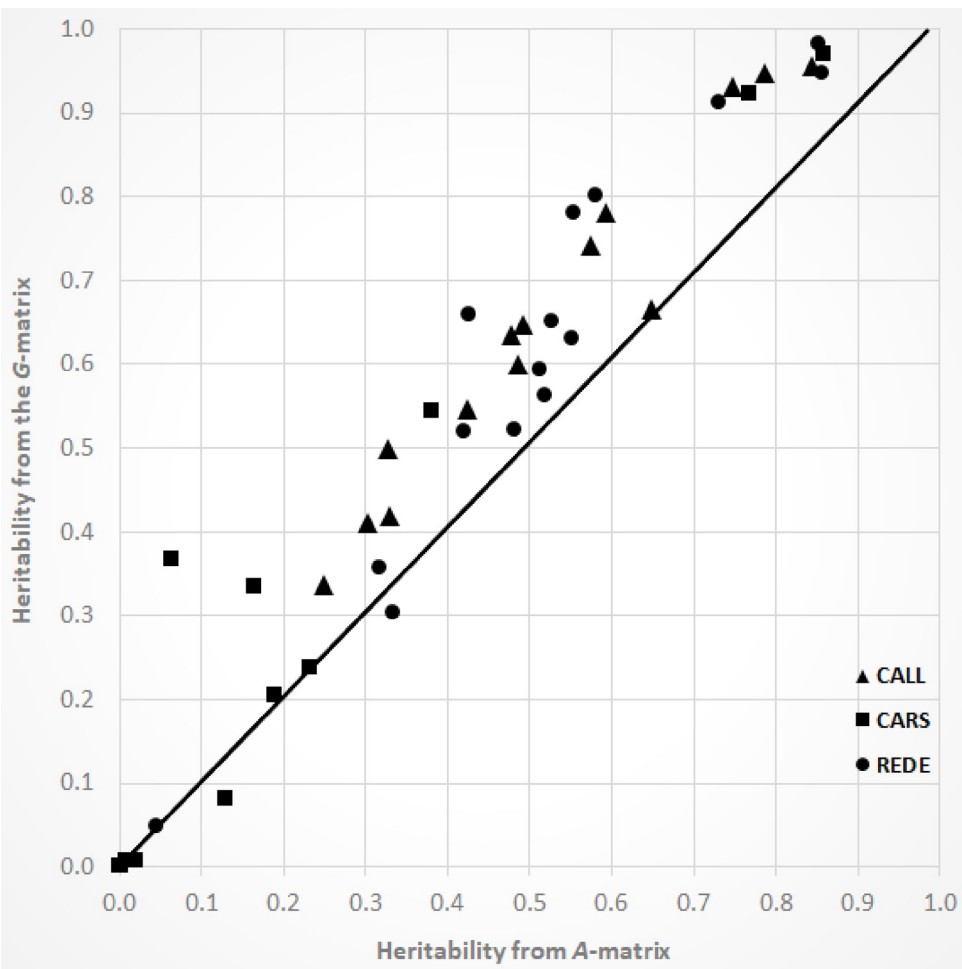

**Fig 2. Scatter plot between estimated narrow-sense heritability estimated from the pedigree- (*A*-matrix) and genomic-based (*G*-matrix) relationship matrices for the 15 studied traits in each of the three white spruces sites.** Abbreviations used for the sites are described in the Table 1.

dispersion along the 1:1 line can be observed in S4 Fig, especially at the CARS site for the correlations between mean drought sensitivity (Sensitivity) and the monoterpene compounds.

Across test sites and relationship matrices, estimates of genetic correlations between DBH and HT were consistently high and positive, ranging between 0.87 and 0.93 (average 0.90). Low to moderate negative or positive correlations were apparent between growth and wood quality traits (WD and MFA) (range: -0.50–0.33), with some inconsistency across sites especially between growth traits and MFA. Generally, consistently low to moderate negative correlations between growth variables and growth resistance (Resistance) were found within and across sites (range: -0.18 –-0.65). However, correlations between growth and Sensitivity traits were high and positive (range: 0.40–0.78) for CALL and REDE, and negative (range: -0.31 –-0.08) for CARS. The correlation between growth traits and $\delta^{13}C$ varied from 0.20 to 0.70. Genetic correlation estimates between growth traits and monoterpene compounds and total monoterpenes were low to moderate. For CALL, correlation coefficients were mostly positive (range: -0.11–0.28), whereas in CARS and REDE low and negative correlations were generally found (range: 0.08 –-0.49) (Fig 3 and S1 Table).

**Table 4. Estimated narrow-sense heritability and their approximate standard error (SE), for each growth, wood quality, drought resilience and chemical traits in the white spruce population.** Heritability estimates were estimated using the pedigree- (*A*-matrix) and genomic-based (*G*-matrix) relationship matrices constructed from all available SNPs (467K). Abbreviations used for the traits and sites are described, respectively, in the text and Table 1.

| Site | CALL | | CARS | | REDE | |
|---|---|---|---|---|---|---|
| Trait | *A*-matrix | *G*-matrix | *A*-matrix | *G*-matrix | *A*-matrix | *G*-matrix |
| HT | 0.747 (0.164) | 0.930 (0.216) | 0.858 (0.264) | 0.971 (0.016) | 0.855 (0.169) | 0.948 (0.012) |
| DBH | 0.592 (0.156) | 0.782 (0.209) | 0.064 (0.212) | 0.368 (0.347) | 0.731 (0.162) | 0.913 (0.219) |
| WD | 0.424 (0.133) | 0.546 (0.185) | 0.166 (0.227) | 0.334 (0.347) | 0.554 (0.158) | 0.781 (0.225) |
| MFA[b] | 0.350 (0.134) | - | 0.233 (0.218) | 0.238 (0.319) | 0.045 (0.105) | 0.049 (0.131) |
| Resistance | 0.249 (0.128) | 0.336 (0.172) | 0.002 (0.003) | 0.002 (0.003) | 0.426 (0.158) | 0.660 (0.227) |
| Sensitivity | 0.326 (0.131) | 0.499 (0.190) | 0.021 (0.211) | 0.008 (0.008) | 0.580 (0.155) | 0.801 (0.225) |
| $\delta^{13}$C | 0.574 (0.150) | 0.743 (0.202) | 0.769 (0.236) | 0.922 (0.355) | 0.853 (0.164) | 0.982 (0.030) |
| α-pinene[b] | 0.491 (0.145) | 0.647 (0.197) | 0.001 (0.002) | 0.001 (0.002) | 0.420 (0.151) | 0.520 (0.202) |
| β-pinene[b] | 0.328 (0.153) | 0.419 (0.211) | *a* | *a* | 0.332 (0.175) | 0.304 (0.218) |
| camphene[b] | 0.843 (0.166) | 0.956 (0.006) | 0.008 (0.008) | 0.008 (0.008) | 0.527 (0.159) | 0.652 (0.216) |
| camphor[b] | 0.302 (0.131) | 0.411 (0.183) | *a* | *a* | 0.551 (0.160) | 0.630 (0.210) |
| myrcene[b] | 0.786 (0.168) | 0.947 (0.215) | 0.380 (0.241) | 0.544 (0.348) | 0.519 (0.169) | 0.563 (0.215) |
| limonene[b] | 0.486 (0.143) | 0.600 (0.189) | 0.189 (0.231) | 0.205 (0.319) | 0.513 (0.162) | 0.594 (0.211) |
| terpinolene[b] | 0.648 (0.197) | 0.666 (0.247) | *a* | *a* | 0.316 (0.151) | 0.358 (0.197) |
| total monoterpene[b] | 0.477 (0.149) | 0.635 (0.199) | 0.129 (0.221) | 0.082 (0.309) | 0.482 (0.159) | 0.522 (0.202) |

[a] Heritability and their approximate standard errors were not estimated at the CARS site due to insufficient phenotypic data.

[b] Logarithmic transformed.

The genetic correlations between the two wood quality traits (WD and MFA) were consistently negative across sites (range: -0.10 –-0.42). Negative correlations were also identified between WD and the drought indices (-0.08 to -0.26 for Resistance, and -0.15 to 0.15 for Sensitivity). In contrast to WD, MFA showed strong and positive correlation values with Resistance (range: 0.39–0.77), while the genetic correlation between MFA and Sensitivity remained low to moderate, and negative (range -0.45–0.19). Genetic correlations between WD and monoterpene compounds and total monoterpenes were generally low and positive, meanwhile MFA also showed generally low but both positive and negative genetic correlation estimates with the various monoterpene compounds.

The adaptability-related Resistance trait showed a low negative correlation with $\delta^{13}$C (range: -0.37 –-0.01) and in general positive correlation with Sensitivity (range: -0.01–0.24). Further, the correlations between the two drought resistance traits varied across sites. For example, the genetic correlations between Resistance and Sensitivity averaged across the two relationship matrices were, -0.80 for REDE, -0.14 for CALL and, with an important variation across the two relationship matrices, 0.19 for CARS. Resistance showed statistically significant low and negative correlations with monoterpenes for CALL, low to moderate positive correlation for CARS, and was low but statistically not significant for REDE. For the Sensitivity and monoterpene associations, strong positive genetic correlations were found for CALL (range: 0.33–0.63), while in REDE, these correlations were mostly non-significant (range: -0.12–0.09, Fig 3 and S1 Table). Correlation estimates between $\delta^{13}$C values and monoterpene compounds and total monoterpenes also varied across sites, with low and negative values for CARS and positive relationships in the remaining sites, although statistically non-significant with relatively large standard errors. Finally, the genetic correlation estimates between monoterpene compounds (including total monoterpenes) were generally moderate to strong, positive and consistent across sites (Fig 3 and S1 Table).

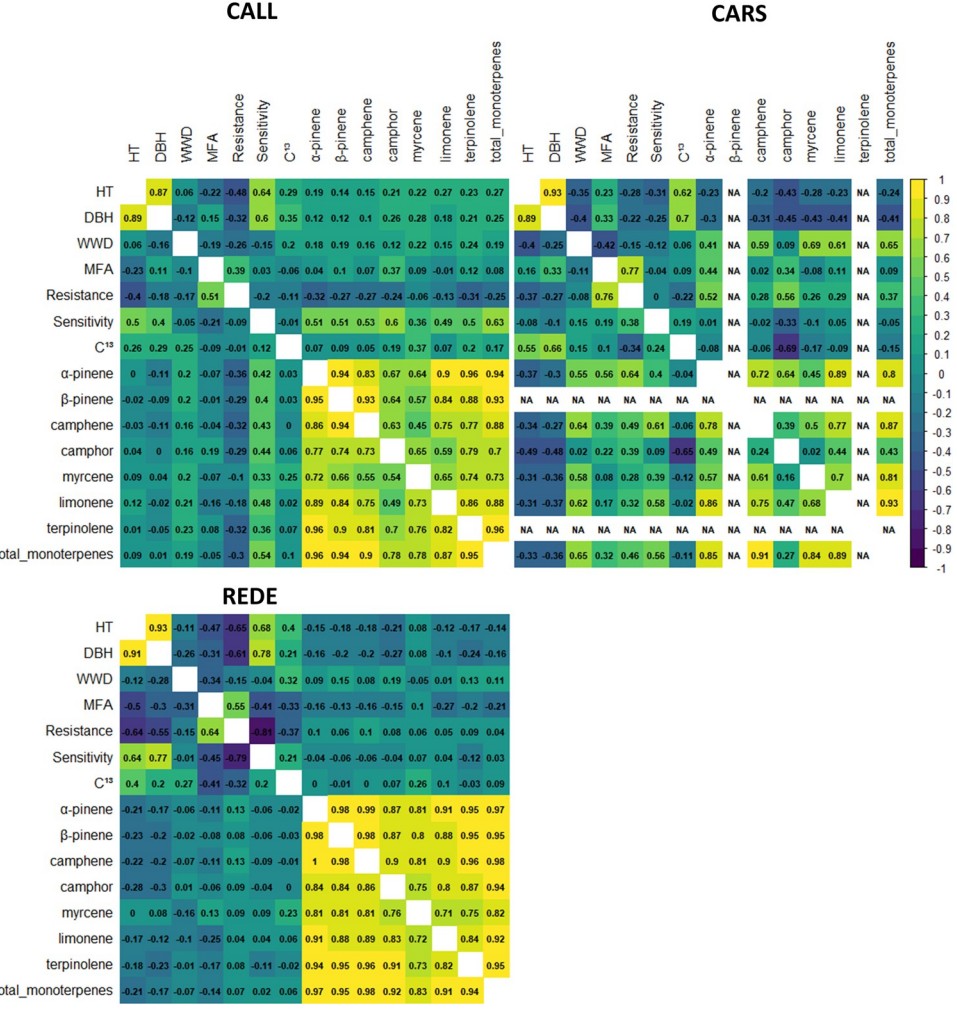

**Fig 3. Estimated genetic correlations between the different traits from the multiple-trait analysis using the pedigree- (*A*-matrix, above diagonal) and genomic-based (*G*-matrix, below diagonal) relationship matrices for the white spruce population.** The genetic correlations are shown in each cell. The color of each individual cell reflects the strength of the genetic correlation, with dark blue and yellow reflecting negative and positive correlations, respectively. Abbreviations used for the traits and sites are described, respectively, in the text and Table 1. **NOTE:** NA = Correlation were not estimated at the CARS site due to insufficient phenotypic data. Transformed data were used for the correlation estimates of MFA and all monoterpene compounds.

### Across sites genetic correlations

On average, across all traits, genetic correlations across sites were similar (in terms of magnitude and direction) regardless of the relationship matrix employed, with only one exception, Sensitivity (S5 Fig). Although the average correlation values among the two multivariate models were similar (0.59 vs. 0.58), the average standard error from the genomic model were double (0.16 vs. 0.32) (S2 Table). Overall, genetic correlations between sites were positive with relatively small standard errors. However, inconsistency was observed, potentially reflecting the climatic conditions between CARS and the other two sites (CALL and REDE) (see Table 1 and discussion below). While average genetic correlation estimates across traits and relationship matrices were strong for the CALL and REDE pair (0.76), the lowest correlations were obtained between the sites CALL and CARS (0.48) and REDE and CARS (0.52), in particular for the growth and MFA traits (Fig 4 and S2 Table).

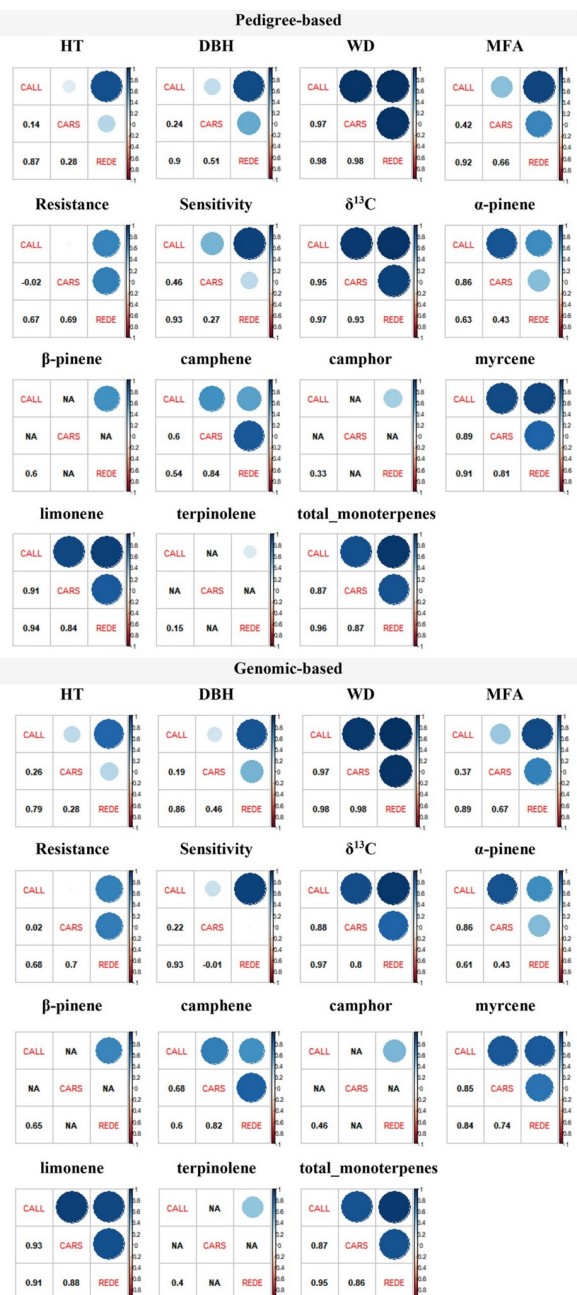

**Fig 4. Estimated genetic correlations between sites for each trait from the multiple-site model using pedigree- (*A*-matrix) and genomic-based (*G*-matrix) relationship matrices for the white spruce population.** The estimated genetic correlations are shown in each cell below the diagonal, and the light to dark blue color of each individual cell above the diagonal reflects the strength of the genetic correlation. Abbreviations used for the traits and sites are described, respectively, in the text and Table 1. **NOTE:** NA = Correlation were not estimated at the CARS site due to insufficient phenotypic data. Transformed data were used for the correlation estimates of MFA and all monoterpene compounds.

For the growth traits (HT and DBH), the average across pedigree- and genomic-based relationship estimates of genetic correlations between pairs of sites was moderate (0.44 and 0.53, respectively). However, as we mentioned above, these correlations were inconsistent for the

pairs of sites that involved CARS, with significantly lower and imprecise (relatively large standard errors) site-to-site genetic correlations. For wood quality traits, genetic correlations among sites were high and consistent across pairs of sites for WD (average 0.98, range 0.97–0.98), while estimates for MFA across sites showed some degree of variability (average 0.66, range: 0.37–0.92) with the lowest correlations (and largest standard errors) also for the pairs involving CARS.

For the adaptability-related drought indices, the genetic correlation among sites for Resistance and Sensitivity, ranged from -0.02 to 0.93, but in general, these estimates were associated with relatively large standard errors, except for Sensitivity between CALL and REDE. Furthermore, significant positive genetic correlations for the WUE related isotopic $\delta^{13}C$ values were found across sites and the two relationship matrices studied (average 0.92, range: 0.80–0.97, Fig 4 and S2 Table).

Genetic correlations for the monoterpene compounds among sites were positive, and ranged from moderate to strong with an average of 0.70 (range: 0.15–0.94). Potentially owing to the smaller sample size ($n < 1,183$, Table 2), the standard errors of the genetic correlations for β-pinene, camphor, and terpinolene between CALL and REDE were larger. Moreover, CARS was not included in these multiple-site analyses of these three compounds as there was insufficient phenotypic data ($n < 30$) available. For total monoterpenes, genetic correlations among sites were positive and strong, and consistent across sites, with an average across the two relationship matrices of 0.90 (range: 0.86–0.96).

## Discussion

Considerable effort have been committed to quantitative genetic analyses of several tree species' growth and wood quality productivity-related traits. While the need to identify adaptability-related trait genotypes grows, less effort has been directed towards the selection of pest and disease resistant trees, and even less for the selection of drought resistant/resilient individuals. Here, we provide a comprehensive quantitative pedigree and genomic analyses of growth, wood quality, drought resilience, and monoterpene traits in a white spruce breeding population. Accurate estimates of narrow-sense heritability and genetic correlation estimates among traits within and across-sites were obtained and are expected to provide valuable information to breed and assist in the selection of resistant/resilient genetic material for increasing productivity and adaptability of future white spruce forests.

### Trait genetic control

Genetic parameters and their function, such as heritability and correlations, play an important role in the selection of parents in a breeding program. However, these values are context dependent, as they depend on the relative contributions of genetic and environmental variations in a specific population, and vary among traits and across measurement ages [52]. While height (HT) heritability estimates (Table 4) showed values somewhat higher than those reported in earlier white spruce studies [10, 14, 16], likely as a result of unintentional sampling artifacts, heritability estimates for diameter (DBH) are consistent with earlier observations in other forest tree species [10, 53]. Although wood density (WD) heritability estimates were comparable to those reported earlier, the pedigree- and genomic-based relationships produced variable results, similar to earlier observations [3, 4, 14]. Microfibril angle (MFA) showed low to moderate genetic control, consistent with results from other white spruce studies [4].

Recent quantitative studies in conifers using population [54], family structure [21], or genomic [55–57] information have used tree-ring traits, such as the short-term index resistance, to analyze the genetic variation and genetic architecture in drought responses. Here, we studied

two short-term indices (i.e., Resistance and Sensitivity) and both produced low to moderate heritability estimates, results similar to those reported by Depardieu et al. [21] in white spruce at a single-site. However, some variation across sites was observed, at CARS for Resistance and Sensitivity, results similar to those reported by Zas et al. [54] who quantified the genetic variation of resilience and resistance indices in two different sites located in central Spain subjected to similar drought events (intensity, timing and duration) in maritime pine (*Pinus pinaster* Ait.). Zas et al. [54] indicated that differences between sites in response to extreme drought events should not be attributed to differences associated with the extreme event itself, but to other microenvironmental factors such as topography, soil depth and stoniness that existed between sites. CARS is the highest elevation site with higher summer precipitation and lower summer temperatures, when compared to the other two white spruce test sites (Table 1). It is therefore plausible that trees in CARS were not exposed to equivalent or severe drought conditions compared to the other two sites to express differences in Resistance and Sensitivity over the same time period (2011–2015).

Resistance to stress is often difficult to measure and depends on a complex network of functional traits at multiple scales [58]. In trees, stable carbon isotope ratio ($\delta^{13}$C) values can be used as an index of integrated long-term water use efficiency (WUE), expressing the ratio of carbon fixed to water lost as related to stomatal function. Moreover, $\delta^{13}$C may serve as a guide for parental selection decisions for seed production, to identify genotypes with contrasting growing strategies, elucidating the underlying mechanisms of complex physiological traits [59], or selecting genotypes for high WUE without compromising yield [60]. In our study, we showed that there is significant potential for selection using $\delta^{13}$C information, as the genetic variation in $\delta^{13}$C was moderate to high, and comparable to earlier reports (Johnsen et al. [61]: *Picea mariana* (0.54); Prasolova et al. [62]: *Araucaria cunninghamii* (range: 0.40–0.72)); however, lower estimates have also been reported (*Pinus pinaster*: Marguerit et al. ([63]; range: 0.23–0.41) and Brendel et al. ([64]; 0.17); *Pinus taeda*: Baltunis et al. ([59]; 0.14 and 0.20 for two sites in Florida and Georgia, respectively)).

Maximizing growth in future climate scenarios with increased pest activity and drought events requires an understanding of the natural variability of quantitative resistance to disease [65] and drought tolerance. In a review on conifers, Kopaczyk et al. [66] indicated that plant secondary metabolites such as terpenes are not involved in vital processes, but may be essential for some conifers to adapt to unfavourable abiotic conditions such as drought stress by increasing levels of constitutive defenses. For instance, total monoterpenes increased significantly in *Pinus sylvestris* (39%) and *Picea abies* (35%) trees under a severe drought relative to that of the control [67]. When *Picea abies* was subjected to water stress, the contents of tricyclene, α-pinene, and camphene were significantly higher than the control trees [68]. Therefore, trees showing resistance to insect attacks or drought events can produce higher levels of secondary chemicals compared with trees susceptible to insects or non-drought stressed trees. In spite of the importance of these compounds in relation to adaptability-related traits, few studies have focused on the genetic control (i.e., heritability estimates) of secondary compounds in forest trees. Hanover [69] reported heritability estimates of five monoterpenes (four of which are included in our study) in *Pinus monticola* ranging from 0.38 to 0.95, with heritability values all within the ranges obtained in our study.

Overall, our results showed that estimates of heritability using the genomic relationship matrix from 467K SNP markers were greater than those estimated using the pedigree relationship matrix (average across traits and sites 0.54 vs. 0.43, respectively; Fig 2), demonstrating that the genetic variance captured depended on whether a pedigree- or genomic-based relationship matrix was used. These results agree with those reported by Tan et al. [70], in *Eucalyptus*, where heritability estimates obtained from genomic information were higher than those

from the pedigree, for both growth and wood quality traits. In contrast, Lenz et al. [71] and Gamal El-Dien et al. [72] found heritability estimates from the genomic relationship matrix lower than those estimated from the pedigree relationship matrix for growth and wood quality traits in *Pice mariana* and *Picea glauca* × *Picea engelmannii*, respectively. However, similar heritability estimates from pedigree and genomic information were obtained for HT in *Picea abies* [73] and HT and MFA in lodgepole pine [29]. These results highlight the differences in genetic parameter estimates that exist between different relationship matrices. The cause of these differences may be attributable to different causes, like different data sets or noise due to uncertainty in the estimates [74]. Interestingly, differences in genetic variance estimates may also exist as a consequence of the fact that pedigree- and genomic-based relationships matrices refer to different base populations, where genomic relationship matrices reflects the genotyped population whereas the pedigree relationships reflects the founders of the pedigreed population [74].

## Relationship among traits

Trait genetic correlations are important for demonstrating their associated genetic responses (how selection on one trait affects the mean and potentially genetic variation in another). This is particularly important for breeders to better understand the interplay between the productivity-related and/or adaptability-related traits. Although higher genetic correlations were observed among growth traits (i.e., DBH and HT) (Fig 3 and S1 Table), such correlation values indicate that selection for any one of these traits alone would give a high correlated response in the other traits, providing an opportunity to efficiently allocate assessment efforts. Our results confirmed previous observations in white spruce [75] and other conifer species [36, 76, 77]. For example, Rweyongeza [75] using progeny trials from the same white spruce series studied here, reported DBH-HT genetic correlation estimates of 0.76 to 0.94 (average 0.85) for age-20 and 30 measurements.

The reported genetic correlations suggest that the selection for rapid growth could result in a small decrease in WD (Fig 3 and S1 Table). Earlier studies in several tree species have shown that genetic correlations between growth traits and WD are negatively correlated, but may also vary with environmental factors (e.g., location, site conditions) [78]. Moreover, different results concerning the relationships between growth rate and WD may be expected, given that WD is a complex trait influenced by many factors [79]. For instance, either negative [13], or no/minor and negative [14] genetic correlation relationships were reported for WD and HT in white spruce. Our results also showed that the genetic correlation between growth traits and MFA depended on the site, as the genetic correlations were low to moderate, as well as negative or positive. A low and negative correlation (-0.31) between MFA and HT was obtained by Park et al. [13] in white spruce; but high and positive or negative correlations (0.71 and -0.52 were reported for MFA and DBH) [80] or moderate correlations (0.40 and 0.39 at age 10 and 25, respectively) [81] in Norway spruce.

Unfavourable results were obtained for the relationship between growth traits and the short-term index Resistance; therefore, selection of larger trees (greater in height and diameter) could result in a decrease in resistance to drought under climate change. Mean drought sensitivity (Sensitivity) also showed an unfavourable relationship with growth traits in CALL and REDE but favourable in CARS, highlighting that differences in relationships can be associated with local environmental conditions [54] (see discussion in the following sub-section). Trujillo-Moya et al. [55] showed that drought resistance was found to be positively correlated with mean annual increment in a 35-year-old Norway spruce provenance test, however, Montwé et al. [82] also showed some contrasting results depending on the origin of the climatic

regions from which 35 lodgepole pine provenances where selected. Montwé et al. [82] also found a trade-off between tolerance to drought and growth only for the most southern (U.S. A.) lodgepole pine population, while the central and southern interior British Columbia (Canada) populations showed an ability to tolerate drought and to maintain comparatively good long-term growth.

Our results showed that faster-growing trees were positively correlated with higher WUE (higher $\delta^{13}C$ values) and this association was strong (Fig 3 and S1 Table). Therefore, these results suggest that $\delta^{13}C$ is a useful criterion for selecting fast growing genotypes with higher WUE. The positive genetic correlation between growth and WUE could arise from several mechanisms. First, the genetic variation in WUE might be driven by the variation in carbon assimilation rate, which in turn, was positively correlated with growth [63]. An alternative interpretation could suggest that the genetic variation in WUE was driven by the variation in stomatal conductance, and taller trees might have lower stomatal conductance due to hydraulic constraint, as found in *Pinus pinaster* (maritime pine) [83]. However, further research is needed to elucidate if our $\delta^{13}C$ findings were driven by the genetic variation in assimilation rate or by stomatal conductance in the studied population, and to explore the causes of the stronger association found between $\delta^{13}C$ and growth traits. Previous studies evaluating field trials did not show the existence of a general trend between growth traits and WUE [61, 63, 84–86]. For instance, in *Picea mariana*, negative [61] and positive [63] correlations between growth and $\delta^{13}C$ were found.

Wood characteristics have been suggested as screening traits for drought sensitivity to identify drought tolerant individuals [87, 88]. Denser wood is typically associated with xylem that is more resistant to hydraulic failure [89]. Our analysis generally showed some unfavourable relationship between WD and Resistance (negative and low correlation), suggesting that average WD values could be a poor predictors of mean drought sensitivity and thus other physiological parameters may be required. Sebastian-Azcona et al. [90] found no differences in cavitation resistance between different provenances of white spruce, which also suggests that other traits such as root water uptake or stomata regulation might have a stronger effect on the inherent differences to drought resistance. George et al. [91], in the genus *Abies*, found that the average ring density had either a negative relationship to resistance or positive relationships to recovery, resilience and relative resilience, as well as no or only weak correlations with different drought events. Other physical properties of wood structure such as MFA may also provide information about tree sensitivity to drought events. Our results suggest that higher MFA values are associated with more drought tolerant trees (higher values of Resistance; positive correlation). Higher MFA may enable the tracheid to bear higher hoop stresses when a tracheid is under high tension given greater resistance against cell collapse during drought events [92]. However, changes in MFA as a reaction to the environment are still poorly understood [92]. In summary, various wood characteristics may be related to drought sensitivity, because the vulnerability of the xylem conduits to hydraulic failure depends on lumen diameter and length as well as on cell wall thickness [55].

In general, our results showed positive genetic correlations between WD and $\delta^{13}C$, with the highest correlations for CALL and REDE (average across the two relationships of 0.23 and 0.30, respectively). Previous studies in *Fagus sylvatica* [93], showed that the phenotypic relationship between WD and $\delta^{13}C$ differed between dry and wet years across sites. For wet years, WD and $\delta^{13}C$ was negatively correlated and, in dry years $\delta^{13}C$ increased with increasing wood density (i.e., positive correlation). Therefore, the higher values observed in the mentioned sites probably are associate with dryer environments. This conclusion can explain the results obtained for CALL and REDE as they are at lower elevation and are drier sites as compared to CARS, located at a higher elevation with relatively moist conditions (Table 1).

Resistance and Sensitivity drought indices were marginally negatively or positively correlated with $\delta^{13}$C, respectively, suggesting that trees most resistant to a drought event have low WUE (i.e., low $\delta^{13}$C values), at least in CARS and REDE, the sites with the highest correlation values (average across relationships, -0.28 and -0.35, respectively). Jucker et al. [86] showed that $\delta^{13}$C values provided a reliable and powerful indicator of drought across a wide range of forest tree species growing in different environmental conditions. As stated above, they did not find enough evidence to suggest that the increase in $\delta^{13}$C was associated with the significant decline in stem growth; however, they showed a clear association between increased $\delta^{13}$C and decreased growth under drought conditions in four sites along a *Picea abies* latitudinal gradient (-31.7% on average, see Fig 2 in Jucker et al. [86]), confirming our results.

There is evidence that terpenes are important components of conifer defenses [94, 95]. It has been shown that some types of stress conditions, such as drought or temperature fluctuation enhance or inhibit the production of terpenes, modify their emission pattern or/and quantity [66]. Thus, the effect of abiotic stress on monoterpenes could explain the different responses across the study sites. For instance, the most resistant trees to drought stress showed low and negative correlations with all the monoterpenes studied at CALL (average correlations across monoterpenes and relationships was -0.25) and positive at CARS (average correlations across monoterpenes and relationships was 0.41). The correlation with the α-pinene concentration (a foliar protectant against *Choristoneura fumiferana* feeding [42]) was the most negative at CALL (-0.34) and the most positive at CARS (0.58). Moreover, it has also been demonstrated that protective compounds produced by plants subjected to biotic stress may enhance their tolerance to abiotic stress [66], the so called "cross-talk" between biotic and abiotic stress responses [96].

Finally, our study also compared the genetic correlation estimates between traits using the classical infinitesimal model from the pedigree information with those estimates from the genomic information. From theory, standard pedigree-based linear models capture expected genetic covariation, whereas marker-based models capture genetic covariation that is marked by SNPs [97]. Therefore, it is expected that for some of these traits, the estimated correlations may depend on the type of information. Our results showed that genome-based correlations generally reaffirm the pedigree-based correlations, but some pairs of traits disagree, either with missing correlations (i.e., the pedigree estimates were higher than those from SNP markers) or excessive correlations (i.e., the SNP markers estimates were higher than those from pedigree) (S4 Fig). Momen et al. [97] highlighted that some care should be taken when interpreting and using genetic parameters estimated via molecular markers, as predictions for complex traits based on pedigree data may differ from those based on SNP data, simply due to chance or other reasons, such as the extent of linkage disequilibrium (LD) between markers and the unknown quantitative trait loci (QTL). To potentially capture parts of the genetic covariance among traits that are not accounted for by either pedigree or genomic information alone, we recommend combining the pedigree and genomic information using the single-step GBLUP approach that combines pedigree and genomic relationship matrix [98] as applied to white spruce [3, 14] and lodgepole pine [2].

## Genetic-environmental correlations

The availability of multi-environmental forest genetics trials makes it feasible to evaluate both the magnitude and importance of the genotype by environment (G×E) interactions [99]. When these interactions are high (genetic correlations < 0.70), breeders must decide whether to select for performance stability and accept a slower rate of population improvement or to develop populations specifically adapted to each environment for gain maximization, however, the latter strategy is usually associated with greater program costs [100].

Despite being in the same breeding region (D1), the climatic conditions varied across the test sites, with the mean annual temperature (MAT) and precipitation (MAP) ranging from 1.3 to 2.9˚C and 442 to 535 mm during the trial period 1986–2019 period, respectively (Table 1). Among the test sites, CARS is higher in elevation, with the highest MAT and lowest mean warmest month temperature (MWMT; i.e., coolest summer), and highest annual precipitation and moisture (see Table 1), while the lower elevation CALL and REDE sites experienced warmer summers, and had lower annual precipitation and moisture index. Overall, these climate differences between CARS and both CALL and REDE sites might explain the high G×E interactions observed for the site-to-site pairs involving CARS, while the climate similarity between CALL and REDE may explain the low G×E interactions between these two sites (Fig 4 and S2 Table), in spite of the large geographic distance between them (Fig 1). It should also be mentioned that CALL and REDE were attacked by white pine weevil (*Pissodes strobi* Peck), a pest that destroys the leading shoot growth.

For HT and DBH, higher G×E interactions were observed for the analyses involving CARS (Fig 4 and S2 Table), suggesting selection for growth at CARS should be considered independently for its unique climate, as well as the absence of damage by white pine weevil. In contrast, for wood quality traits, such as WD and MFA, our results indicated a neglectable G×E effect. Similar results have previously been reported for several conifer species [2, 101–105]. These studies revealed that G×E interaction for WD is not very important (lodgepole pine: Ukrainetz and Mansfield [2] > 0.78; *Pinus radiata*: Baltunis et al. [104] > 0.74 and Gapare et al. [105] > 0.70; Chen et al. [102] > 0.74; *Pinus taeda*: McKeand et al. [103] = 0.77). Furthermore, although we identified G×E effect at the higher elevation CARS site, previous studies showed little G×E interactions for MFA. For instance, Baltunis et al. [101] showed a mean Type B genetic correlation [106] of 0.87 in *Pinus radiata* families from two second-generation progeny trials. In two large open-pollinated progeny trials of Norway spruce, established in southern Sweden, Chen et al. [102] also observed high Type B genetic correlations (0.85) for MFA.

The importance of examining the genetic variation in drought resilience across a range of extreme climate events and across sites has been emphasized [54]. However, to date, most studies have focused on single test sites [21, 55, 56]. Our findings showed high variability in genetic correlations between study sites with relatively large standard errors for the drought response indices, except for mean drought sensitivity (Sensitivity) between CALL and REDE. We have concluded that selection for drought resistant genotypes can only be made for sites with similar climate indices (such as the CALL and REDE in this study). How forests and trees react to drought is complex and varies across stands, sites, regions, and continents depending on multiple factors including climate conditions [107]. Moreover, as we mentioned before, these differences are likely due to other microenvironmental factors that existed between these three white spruce test sites, as indicated by Zas et al. [54].

The small G×E interaction reported in our work for $\delta^{13}C$ are in agreement, generally, with other previous conifer studies [61, 108]. Johnsen et al. [61] showed no evidence for a G×E interaction for foliar carbon isotope discrimination in *Picea mariana*. Guy and Holowachuk [108] reported no significant G×E interactions for $\delta^{13}C$ in 10 lodgepole pine provenances tested on three sites in British Columbia (Canada) with contrasting soil moisture and climate. However, Baltunis et al. [59] observed a lower value of Type B total genetic correlation (0.64) in 1,000 *Pinus taeda* cloned full-sib families tested on two contrasting sites. Cregg et al. [109] also observed strong G×E interaction for stable carbon isotope discrimination in mature *Pinus ponderosa* at two contrasting locations in the Great Plains (USA), caused by growth phenology variation among seed sources. Information on G×E interaction for $\delta^{13}C$ of white spruce field trials is extremely limited.

Finally, in agreement with two previous studies [110, 111] we observed, with only a few exceptions, low G×E interactions in the monoterpene compounds and total monoterpenes. Few studies have investigated G×E interaction of monoterpenes, probably, as stated by Ott et al. [111], due to the need for spatially replicated field trials of trees with known pedigree that are of an appropriate age for biotic challenges of interest. Hanover [110] showed that five cortical monoterpene concentrations (four used in this study) were quite stable across three clonal *Pinus monticola* trials established in contrasting sites in Idaho (USA). Ott et al. [111] also found that only a few monoterpene compounds from phloem tissue showed significant family × environment interactions in two OP progeny trials of lodgepole pine established in north central British Columbia. Based on these results, we can conclude that monoterpene compounds found in needles are relatively independent of climate and site characteristics, at least, within the studied D1 white spruce breeding region.

## Implications for white spruce breeding

The ultimate goal of forest tree breeding and testing programs is to evaluate parents and their offspring across multiple sites, and for a reasonable duration to make reliable selection for the next breeding cycle. These efforts allow for the establishment of seed orchards for the reliable and abundant production of improved seed needed for reforestation programs today, and typically for the life of the orchard. Here, we tested and genotyped 80 of 150 families from a white spruce breeding population planted on three sites within one breeding zone for multiple traits including growth, wood quality, drought resistance, and chemical compounds associated with biotic and abiotic resistance using genomic (SNPs) and pedigree derived relationships. We compared the target attributes' genetic parameters (estimates of heritability, genetic correlation, and G × E interactions) using the two relationship methods to reach a reliable conclusion on selecting the appropriate genetic evaluation method as well as understanding the interplay among the selection traits to allow for more rapid evaluation without compromising the selection accuracy. The choice of selection attributes is of vital importance considering the time and effort needed for phenotypic traits evaluation, understand the correlated responses among the target traits, and finally the G × E interactions. In this regard, a number of findings can be made based on the results of this study aiming at improving white spruce breeding efforts challenged with the need to increase the scope of selections attributes and target deployment environments. The key findings include the following: a) use of $\delta^{13}C$ as a relatively easy to measure trait and is an excellent proxy to WUE and growth rate, b) use of secondary chemical compounds (monoterpenes) as an indicator of a selected trees propensity to show insect and/or drought resistance, c) while the **G**-matrix provided better genetic parameter estimates than the **A**-matrix, the inconclusiveness of the former in some cases indicated that a blind approach (i.e., single-step GBLUP approach) of these relationships would be best, d) the existence of positive and negative genetic correlations among the studied traits cannot be overlooked during selection, e) the unfavourable relationship between growth and wood quality traits with drought resistance indices (negative correlations), indicating the importance of proper trait(s) choice for selecting under expected increasing drought environment with climate change, f) the value of chemical compounds "cross-talk" as an indicator for tolerance to biotic and abiotic stress, g) the magnitude and trajectory of G × E interaction as it determines the selection strategy (i.e., specialists vs. generalists), which is essential for seed orchards establishment, and h) the value of multiple site testing, especially for drought resistance, as variability among testing sites provide insight into site differences even if they are within one breeding zone. We believe that the lessons learned from this study will provide valuable information in the future selection and breeding of the white spruce population in Alberta and elsewhere.

## Supporting information

**S1 Fig. Annual variation in average basal area increment (BAI) of the open-pollinated white spruce families for the period 1995–2016 at each of the three test sites.** The red dashed line represents the year of the drought event and the green shadowed area represents the pre-drought period considered to calculate the Resistance index.
(DOCX)

**S2 Fig. Density distribution for the studied traits in white spruce in each of the three test sites.** Logarithmic transformations were applied to MFA and all monoterpene compounds to improve data normality. Abbreviations used for the traits and sites are described, respectively, in the text and Table 1.
(DOCX)

**S3 Fig. Pedigree and genomic relationships.** Distribution of the number of pairwise additive relationships (excluding the diagonal elements) from the pedigree (after pedigree correction, left) and genomic (right) relationship matrices. Note that *y*-axis (Frequency) were cut at 40,000 (*A*-matrix, out of 2,343,490) and at 10,000 (*G*-matrix, out of 1,555,212) in order to more clearly visualize the differences between relationship matrices.
(DOCX)

**S4 Fig. Scatter plot between estimated genetic correlation between pairs of traits from the pedigree- (*A*-matrix) and genomic-based (*G*-matrix) relationship matrices in each of the three white spruce sites.** Abbreviations used for the sites are described in the Table 1.
(DOCX)

**S5 Fig. Scatter plot between estimated genetic correlation between pairs of sites from the pedigree- (*A*-matrix) and genomic-based (*G*-matrix) relationship matrices in each of the 15 assessed traits in white spruce.** Abbreviations used for the traits are described in the text.
(DOCX)

**S1 Table. Estimated genetic correlations (and approximate standard errors) between the different traits from the multiple-trait analysis using the pedigree- (*A*-matrix, above diagonal) and genomic-based (*G*-matrix, below diagonal) relationship matrices for white spruce in each of the three test sites.** Abbreviations used for the traits and sites are described, respectively, in the text and Table 1.
(DOCX)

**S2 Table. Estimated genetic correlations (and approximate standard errors) between the different sites from the multiple-site analysis using the pedigree- (*A*-matrix, above diagonal) and genomic-based (*G*-matrix, below diagonal) relationship matrices for white spruce in each of the three test sites.** Abbreviations used for the traits and sites are described, respectively, in the text and Table 1.
(DOCX)

**S1 Text. Chemical analysis.**
(DOCX)

## Acknowledgments

We would like to acknowledge the RES-FOR staff that collected and prepared the many white spruce samples for this research: Laura Vehring, Pablo Chung, Jillian Dyck, Sarah Suzuk,

Kristie Bui, Chris Arbter, Rob Johnstone, Jesse Shirton, Arial Eatherton, Calvin Jensen and Michael Thomson.

## Author Contributions

**Conceptualization:** Nadir Erbilgin, Barb R. Thomas, Yousry A. El-Kassaby.

**Data curation:** Eduardo P. Cappa, Jennifer G. Klutsch, Jaime Sebastian-Azcona, Blaise Ratcliffe, Xiaojing Wei, Letitia Da Ros, Charles Chen, Andy Benowicz.

**Formal analysis:** Eduardo P. Cappa.

**Funding acquisition:** Nadir Erbilgin, Barb R. Thomas, Yousry A. El-Kassaby.

**Investigation:** Eduardo P. Cappa, Jennifer G. Klutsch, Jaime Sebastian-Azcona, Blaise Ratcliffe, Xiaojing Wei, Letitia Da Ros.

**Methodology:** Eduardo P. Cappa, Blaise Ratcliffe.

**Project administration:** Barb R. Thomas.

**Resources:** Charles Chen, Andy Benowicz, Shane Sadoway, Shawn D. Mansfield, Nadir Erbilgin, Barb R. Thomas, Yousry A. El-Kassaby.

**Software:** Eduardo P. Cappa.

**Supervision:** Shawn D. Mansfield, Nadir Erbilgin, Barb R. Thomas, Yousry A. El-Kassaby.

**Writing – original draft:** Eduardo P. Cappa, Jennifer G. Klutsch, Jaime Sebastian-Azcona, Blaise Ratcliffe, Xiaojing Wei.

**Writing – review & editing:** Eduardo P. Cappa, Jennifer G. Klutsch, Jaime Sebastian-Azcona, Blaise Ratcliffe, Xiaojing Wei, Letitia Da Ros, Yang Liu, Charles Chen, Andy Benowicz, Shane Sadoway, Shawn D. Mansfield, Nadir Erbilgin, Barb R. Thomas, Yousry A. El-Kassaby.

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
