## [Decision Letter · Decision Letter 0]

22 Oct 2021

PONE-D-21-25537Quantitative pedigree and genomic analysis of productivity and climate-adaptability traits in white sprucePLOS ONE

Dear Dr. Cappa,

Thank you for submitting your manuscript to PLOS ONE. After careful consideration, we feel that it has merit but does not fully meet PLOS ONE’s publication criteria as it currently stands. Therefore, we invite you to submit a revised version of the manuscript that addresses the points raised during the review process. Please submit your revised manuscript by Dec 06 2021 11:59PM. If you will need more time than this to complete your revisions, please reply to this message or contact the journal office at plosone@plos.org. Please include the following items when submitting your revised manuscript:A rebuttal letter that responds to each point raised by the academic editor and reviewer(s). You should upload this letter as a separate file labeled 'Response to Reviewers'.A marked-up copy of your manuscript that highlights changes made to the original version. You should upload this as a separate file labeled 'Revised Manuscript with Track Changes'.An unmarked version of your revised paper without tracked changes. You should upload this as a separate file labeled 'Manuscript'.

We look forward to receiving your revised manuscript.

Kind regards,

Ricardo Alia, PhD

Academic Editor

PLOS ONE

2. Please upload a new copy of Figures 3 and 4 as the details are not clear. Please follow the link for more information:  http://blogs.plos.org/everyone/2011/05/10/how-to-check-your-manuscript-image-quality-in-editorial-manager/ and and this link for our full figure guidelines http://journals.plos.org/plosone/s/figures.

“This work was completed utilizing the High-Performance Computing Center facilities of Oklahoma State University (NSF MRI-1531128), and also in part by the Extreme Science Foundation Environment (XSEDE, NSF ACI-1548562). Specifically, it used the Bridges and the Bridges2 system, which was supported, respectively, by NSF award number ACI-1548562 and ACI-1445606 at the Pittsburgh Supercomputing Center (PSC). The allocation is under the MCB180177 of Charles Chen. We also would like to acknowledge the RES-FOR staff that collected and prepared the many white spruce samples for this research:  Laura Vehring, Pablo Chung, Jillian Dyck, Sarah Suzuk, Kristie Bui, Chris Arbter, Rob Johnstone, Jesse Shirton, Arial Eatherton, Calvin Jensen and Michael Thomson.”

“We acknowledge cash funding for this research from Genome Canada, Genome Alberta through Alberta Economic Trade and Development, Genome British Columbia, the University of Alberta, and the University of Calgary. Further cash funding has been provided by Alberta Innovates BioSolutions, Forest Resource Improvement Association of Alberta, and the Forest Resource Improvement Program through West Fraser Ltd. (Blue Ridge Lumber and Hinton Wood Products) and Weyerhaeuser Timberlands (Grande Prairie and Pembina).  In-kind funding has been provided by Alberta Agriculture and Forestry, Blue Ridge Lumber West Fraser, Weyerhaeuser Timberlands Grande Prairie, and the Thomas, Wishart, and Erbilgin labs in support of the Resilient Forests (RES-FOR): Climate, Pests & Policy – Genomic Applications project.”

6. We note that Figure 1  in your submission contain [map/satellite] images which may be copyrighted. All PLOS content is published under the Creative Commons Attribution License (CC BY 4.0), which means that the manuscript, images, and Supporting Information files will be freely available online, and any third party is permitted to access, download, copy, distribute, and use these materials in any way, even commercially, with proper attribution. For these reasons, we cannot publish previously copyrighted maps or satellite images created using proprietary data, such as Google software (Google Maps, Street View, and Earth). For more information, see our copyright guidelines: http://journals.plos.org/plosone/s/licenses-and-copyright.

Additional Editor Comments:

The paper present an interesting analysis of a breeding popoulation of white Spruce. However, as raised by the reiewer 2, the paper present some aspects that can be improved, in the objectives, and discussion of the resutls. I suggest to read carefully the comments to produce a revised version of the manuscript, as there are different suggestion that can be easily addressed in the revised manuscript.

Reviewers' comments:

Reviewer's Responses to Questions

**Comments to the Author**

1. Is the manuscript technically sound, and do the data support the conclusions?

Reviewer #1: Yes

Reviewer #2: Yes

2. Has the statistical analysis been performed appropriately and rigorously? 

Reviewer #1: No

Reviewer #2: Yes

3. Have the authors made all data underlying the findings in their manuscript fully available?

Reviewer #1: No

Reviewer #2: Yes

4. Is the manuscript presented in an intelligible fashion and written in standard English?

Reviewer #1: Yes

Reviewer #2: Yes

5. Review Comments to the Author

Reviewer #1: Cappa et al. evaluated the genetic parameters of productivity and climate-adaptability traits in white spruce. In forest tree species, this maybe is the first paper to introduced genetic variation and genetic parameters of productivity-, and adaptability-related traits as well as chemical defence compounds (monoterpenes), thus, the paper is important for a white spruce breeding program, especially under a quick climate change in boreal forests. In theory, the results should be reliable based on 80 half-sib families and 1540 individuals. But I am not confident about their sampling strategy base on such high heritabilities for tree height and DBH. Thus, several generally comments may need to address before being accepted:

1) I was wondering why the author did not present the results by joint-site analysis by estimating GXE term as a standard way?

2) Based on the paper (Rweyongeza 2016), would you present the results for tree height or DBH using the whole families (n=150)? Will this change the heritability and genetic correlation?

3) How accurate of your GBS data? did you estimate the accuracy of imputation? Also for GBS data, the different methods to estimate G matrix could produce a large effect for estimating relationships (Dodds et al., 2015).

Some minor comments:

L41: What is the genotyping platform used?

L63: Any citations?

L130 why do you select 80 families based on low, average and high-class height? Not random?

L132, is 34 potential trees in the same trial series?

Interesting, there is no overlap between the 19 families and 80 families if you also selected families from high-class heights.

L136: Could you add the number of trees selected in Table 1?

L166: Why did you select four years, not two, three or five before the drought event? It would be good to show the trajectory of BAI for each trials as the paper (Depardieu et al., 2020) et al. 2021) did in New phytologist.

L202. It would be better if the distribution information could be shown in the supplementary file.

L204, please remove Tan et al. 2016 in here

L213, where is median?

L225. a maximum missing data proportion for individual or locus? a minor allele count of one? Why?

L224 and 226 the description was not sound logistic.

For example, after maximum site depth <=70, a minor allele count of for each locus, then it would be a maximum missing data proportion of 30%?

L236 if the parental trees are from 99 to 204, will this reduce the accuracy of those additional parents as it may only have one progeny for each parent?

L247: sigma^2 is the additive genetic variance

L248 σ^2*I?

L258: [′ | ⋯ | ′] is a matrix?

L271: Based on your genotypic data is from GBS genotyping platform, have you considered using other methods, such as Godds’ method or others to estimate pair-wise relationships (Dodds et al., 2015)? In this study, what imputation method did you use before you estimate the G matrix? As we know, GBS is quite sensitive for estimating relationships. Have you considered this?

L356 the number of individuals selected in each trial is important.

L419: red and dark blue?

L466: Did you test your assumption for such higher heritability estimates for tree height and DBH? Based on the paper (Rweyoneze 2016), all families were measured at least in two trials in many years.

L467-468. I think you need to discuss more for such high heritability, not comparing the heritability between height and DBH. Only a small difference, I don’t think it is abnormal or need to discuss between height and DBH.

L472-483 WD is lower than height and DBH? It is interesting for me.

What about other traits used in (Depardieu et al., 2020)? E.g. growth resilience, growth recovery, growth relative resilence?

L513: 2008)

L536: Fig 3 and Table S1?

L597: the results from Hannrup and Lenz et al. should be introduced together.

L672: delete Cappa et al. 2012

L697: what about Picea abies (Chen et al. 2014)?

L713: what are microenvironmental factors?

If the site was not under drought stress, it may be difficult to accurately estimate the value and genetic parameters.

Depardieu C, Girardin MP, Nadeau S, Lenz P, Bousquet J, Isabel N. 2020. Adaptive genetic variation to drought in a widely distributed conifer suggests a potential for increasing forest resilience in a drying climate. New Phytologist.

Dodds KG, McEwan JC, Brauning R, Anderson RM, van Stijn TC, Kristjánsson T, Clarke SM. 2015. Construction of relatedness matrices using genotyping-by-sequencing data. Bmc Genomics 16(1): 1047.

Reviewer #2: This manuscript shows quantitative genetic parameters in a subsample of a White spruce breeding population in the province of Alberta (Canada), estimated using both pedigree and genomic based relationships. The traits addressed here were of productive (growth and wood characteristics) but also of adaptive meaning (drought and pest resistance) in order to provide information for decision making within the breeding program for more productive and resilient spruce forests in the region.

This is a relevant manuscript for several reasons. Firstly, part of the novelty is because the genomic implementation in an operational tree breeding program. Although is not the first report about genetic estimations upon genomic predictions (see for example Ukrainetz & Mansfield 2020, Tree Genet & Genomes), it fits within the first stages of application of this new methodology in forest species such that new results and comparisons are much needed. Second, the array of traits is pretty much relevant and costly to get including x-ray wood density estimations, isotopic discrimination in timber tissues, dendrochronological parameters and chemical defenses. Thus, this work builds a unique dataset within the forest breeding programs field in order to address questions related to both breeding and ecological and evolutionary matters. Third, three environments have been tested, such that plasticity effects and genotype by environment interactions patterns can be disentangled. Finally, methodology and specifically the statistical analyses are sound and robust, given that from my perspective authors have deployed most modern and robust methodologies for quantitative genetics analyses in forest species, including the always-welcomed spatial corrections and proper linear mixed models.

Overall, the manuscript is well written and language is clear and stylish, it takes advantage of a relevant experimental design, implements an accurate methodology, introduces the topic properly and highlights the importance of addressing adaptive traits in forest breeding programs under current global change, shows materials and methods properly and take advantage of proper and updated bibliography. The manuscript is relevant for forest breeding science and its methodology is convincing. As consequence, I believe that the manuscript may be suitable for publication in ‘PlosONE’.

However, I am missing a more structured message, with a main message in front together with some other secondary messages. I have the feeling that the manuscript misses the opportunity to build a more meaningful message of interest for breeders and for even ecologists and evolutionary biologists interested in forest trees. Following are the symptoms I found within the manuscript which shows the lack of a particular and robust message:

1. The title is ambiguous, and although it clearly shows what has been done it does not show what has been obtained.

2. At the end of the introduction, no hypotheses are shown but authors point to the following goal: “L116-118. We studied 15 growth, wood quality, drought resilience, and defense and drought stress chemical traits (monoterpenes), and estimated their quantitative genetic parameters (including heritability and genetic correlations) within and across-sites”. Here, I do not agree that estimating genetic parameters per se is a goal for a scientific paper. Furthermore, in the same section author’s state: “L119-121. The results of this study would provide critical information for the identification and selection of genetic material for the production of productive, healthy, and resilient white spruce future forests”. Hence, if the critical information has been produced, why is not discussed in the current manuscript in terms of future strategies for the breeding program?

3. Although a relevant effort has been done in order to justify results and to put them in context in terms of current bibliography, large parts of the discussion are mainly comparisons of estimated genetic parameters with the ones obtained in other breeding programs and even for other species (e.g. L.545-560; L586-592; L.594-597). I believe that it is relevant to highlight that genetic parameters, as for instance heritability, are context dependent, as they depend on specific populations and specific traits in a specific time or environmental context. Thus, heritability comparisons with other populations or even other species and under different environmental conditions, although can be relatively useful to put in context some results, they have a limited relevance in order to provide meaning to a discussion.

Hence, given that the authors are not clearly showing a main message, the manuscript looks like unfocused, novelty is not clearly stated (too many comparisons with former works such that it looks like that everything has been done before) and may end in reduced interest for a potential audience. Furthermore, the results section is quite difficult to follow given the big list of genetic parameters estimated in 3 different sites for 15 different traits, including significance and standard errors.

My recommendation to the Editor is to ask for a major revision such that the manuscript can be rewritten to become more centered in specific messages, to attain higher scientific quality and become more helpful for potential readers.

Following I am attaching some thoughts and ideas with the aim to inspire authors and to let them know where is the gap that in my opinion exists:

1. Discussion is too long and does not help to center the message. For instance, some paragraphs as L.463-471 are not novel results at all in forest breeding, thus I believe that it should be removed in order to make proper room for the relevant messages. Instead, paragraphs as L.598-614 are a good example of a more message-centered discussion that may be a reference to rewrite discussion.

2. A specific goal should be highlighted from the very beginning, whether it is to show a discussion about future breeding strategies within the breeding program upon current results, or to center the discussion in evolutionary and physiological trade-offs among adaptive and growth traits or even a combination of both perspectives. From my point of view, the traits measured here have a pretty big room for physiological and evolutionary discussion what is enhanced by the selection of a wide random population with half-sibs structure tested in 3 sites.

3. Other potential focus of discussion for the manuscript in the comparison between pedigree and genomic based relationships estimations, given that I have the feeling that it has not been properly discussed even if the topic is shown in the title.

Finally, following I point specific mistakes, advices or concerns:

L.63-65. References are missing.

L.88-90. I believe this statement is too risky. Authors should be more explicit in terms of what actually has not been studied before.

L.107-109. Again, I believe this statement is too risky. Authors should be more explicit in terms of what actually has not been reported before.

>L.117. Here it should be shown the main goals, and also some hypothesis if needed.

L.129. “Testing population” does mean the original breeding population that has been subsampled? State it clear.

L.192. If I am not wrong it should be -40ºC, right?

L.242-243. Which is the reason for accounting for the provenance effect as a fixed effect in the model? Based on the goals that seem to be sought, should not be better to concentrate the whole genetic variation in the family effect for breeding purposes?

L.245. Please list the whole list of random effects for clarity.

L.263. A further explanation about ‘A’ is needed.

L.267. A further explanation about ‘I’ is needed.

L.452-455. My impression is that in forest breeding programs over the last 20 years the most relevant target traits after productivity are pest and disease resistance. Although I am agree that in terms of drought resistance we still are in very preliminary stages, I believe authors should rewrite this sentence or make a further rationale justifying the statement.

L.459-461. If “valuable information is expected to be provided to bred and assist in selection”, why authors do not discuss this information in terms of breeding strategies?

L.459. “Breeds” should be “breed”

L.526-527. I am not totally sure about this statement. Moreira, Sampedro, Zas and collaborators for instance have amply published about this question.

L.534. “breeders” instead of “breeds”.

L.570. Delete “his studied”?

L.622. “..in dry years δ13C the correlation increased..” something is wrong here.

L.628. “resistant” instead of “resistance”.

L.665-666. Why authors did not applied same methodology they advise?

L.672. Citation “Cappa et al. 2012” is repeated twice.

L.683. Delete “in” in “lower in annual precipitation…”?

L.720-722. Baltunis et al (2008) showed low genetic correlation and strong G×E as consequence. Hence, this citation here together with other studies which found no significant G×E like Guy and Holowachuk (2001) does not match. It should be together Cregg et al. (2000) who observed strong G×E.

Table 2. Min. and Max. values for ‘HT are in the wrong units.

Fig. 2. This is not the proper Figure. Actually, Fig.2 in main text is exactly the same than S1 Fig. which show genetic correlations and not heritabilities as it should be.

6. PLOS authors have the option to publish the peer review history of their article (what does this mean?). If published, this will include your full peer review and any attached files.

Reviewer #1: **Yes: **Zhiqiang Chen

Reviewer #2: No

---

## [Author Response · Author response to Decision Letter 0]

5 Dec 2021

Responses to Reviewer #1

“Cappa et al. evaluated the genetic parameters of productivity and climate-adaptability traits in white spruce. In forest tree species, this maybe is the first paper to introduced genetic variation and genetic parameters of productivity-, and adaptability-related traits as well as chemical defence compounds (monoterpenes), thus, the paper is important for a white spruce breeding program, especially under a quick climate change in boreal forests. In theory, the results should be reliable based on 80 half-sib families and 1540 individuals. But I am not confident about their sampling strategy base on such high heritabilities for tree height and DBH. Thus, several generally comments may need to address before being accepted:”

“1) I was wondering why the author did not present the results by joint-site analysis by estimating GXE term as a standard way?”

We studied the magnitude and importance of the genotype by environment (G×E) interactions using a multiple-site individual-tree mixed model for each trait with an unstructured variance-covariance matrix of the G×E effects, i.e., with different covariances between any two sites. The reviewer recommended the use a joint-site analysis using a multiple-trait individual-tree mixed model with a G×E term, i.e., with a single G×E variance across site for each trait. However, we would like to note that while average genetic correlation estimates across traits and relationship matrices were high for the CALL and REDE pair (0.76), lower correlations were estimated between the sites CALL and CARS (0.48) and REDE and CARS (0.52), in particular for the growth and MFA traits. Therefore, we believe that the model used with an unstructured variance-covariance matrix for the G×E effects is the best analysis given that we can capture these differences in the magnitude of the G×E interactions between the different pairs of sites, a unique G×E term would obtain a simple average of the pair sites correlations, i.e., an average of the G×E interactions.

“2) Based on the paper (Rweyongeza 2016), would you present the results for tree height or DBH using the whole families (n=150)? Will this change the heritability and genetic correlation?”

Following this comment, we performed a pedigree-based (ABLUP) and pedigree and genomic-based (single-step BLUP approach, HBLUP) preliminary analyses, using all the available data for these three white spruce test sites (i.e., 150 families and 15,072 trees) for diameter at breast height (DBH), and height (HT). The following Table shows the variance components and narrow-sense heritability estimates for each site-trait combination using ABLUP and HBLUP analyses for all available dataset (150 families). 

Site

 Trait

 ABLUP HBLUP

 Additive Residual Heritability Additive Residual Heritability

CALL DBH 3.72 8.56 0.30 3.79 8.51 0.31

CARS DBH 1.42 10.91 0.11 1.63 10.74 0.13

REDE DBH 4.48 8.71 0.34 4.57 8.63 0.35

CALL HT 1.77 2.02 0.47 1.79 2.00 0.47

CARS HT 0.53 1.51 0.26 0.57 1.48 0.28

REDE HT 2.40 2.47 0.49 2.39 2.49 0.49

These heritability estimates are lower than those already reported in Table 3 for the 80 RES-FOR families sampled, with the exception of DBH at CARS. The following Table shows the genetic correlations across-trait and across-sites using ABLUP (above diagonal) and HBLUP (below diagonal) for all available datasets (150 families). Trait-to-trait genetic correlations for each test site are similar to those already estimated with 80 families for CALL and REDE, while CARS showed some differences (~ 0.54 vs ~ 0.91 calculated from the 150 and 80 families respectively). Site-to-site genetic correlations are comparable between both datasets (i.e., 80 vs 150 families). Thus, we feel that our analyses present the same picture and additional analyses (see Tables) is not warranted.

Site Trait CALL CARS REDE CALL CARS REDE

 DBH DBH DBH HT HT HT

CALL DBH 0.32 0.95 0.87 0.27 0.79

CARS DBH 0.33 0.25 0.15 0.54 0.05

REDE DBH 0.95 0.25 0.83 0.10 0.87

CALL HT 0.88 0.17 0.83 0.25 0.84

CARS HT 0.28 0.53 0.12 0.25 0.11

REDE HT 0.79 0.07 0.87 0.84 0.11 

“3) How accurate of your GBS data? did you estimate the accuracy of imputation? Also for GBS data, the different methods to estimate G matrix could produce a large effect for estimating relationships (Dodds et al., 2015).”

We have tested the accuracy of the implemented imputation in spruce by cross validation and feel confident with the SNP file used. Please see details of imputation evaluation in the following manuscript: Gamal El-Dien O, Ratcliffe B, Klapste J, Chen C, Porth I, El-Kassaby YA. Prediction accuracies for growth and wood attributes of interior spruce in space using genotyping-by-sequencing. BMC Genomics. 2015; 16: 370. doi:10.1186/s12864-015-1597-y.

We appreciate the Dodds et al. 2015 findings, but with the number of SNPs used (> 460K), we feel that the subtle differences among algorithms would detract us from addressing our stated objectives (see also answer below in the minor comment “L271: Based on your genotypic …”).

Some minor comments:

“L41: What is the genotyping platform used?” 

The genotyping platform used was added.

“L63: Any citations?” 

As example, three references were added.

“L130 why do you select 80 families based on low, average and high-class height? Not random?”

While the sampling appears to be “non-random”, the families were ranked based on their height performance and we “randomly” selected representatives of all performance classes to capture the maximum variation present in the tested population. 

 “L132, is 34 potential trees in the same trial series?

Interesting, there is no overlap between the 19 families and 80 families if you also selected families from high-class heights.”. 

First, we regret we have not expressed ourselves clearly enough about the number of potential forward selected trees sampled in this population. As we already stated in the original manuscript (Lines 133-134), the number of trees from the additional 19 families not included in the original 80 sampled families is 34. However, the total original number of potential forward selected trees samples is 142: 108 trees from 33 families overlap with the original 80 sampled families, and 34 trees from 19 families that did not overlap with the 80 sampled families. Therefore, we are sorry for not expressing ourselves clearly when we reported the number of total potential forward selected trees from the original sampled trees (1,625), this number was corrected and is as “142”, replacing the incorrect value of “34”. These original 142 potential forward trees come from the same three test sites (45 from CALL, 18 from CARS, and 71 from REDE). Therefore, we have corrected the previous sentence which now reads: “… and four progeny for CARS site (n = 1,483). An additional 142 potential forward selected trees, previously identified in the three progeny trials and based on height breeding values, were also included for sequencing. From these 142 forward selected trees, 34 trees were from an additional 19 families, resulting in a total of 1,625 trees from 99 families.”.

Now, the number of trees (108 out of 142) and families (33 out of 80) included in the original 80 sampled families is significantly higher than the trees (34) and families (19) with no overlap with originally sampled trees and families. 

“L136: Could you add the number of trees selected in Table 1?”

The number of original trees selected by test site was added to Table 1.

“L166: Why did you select four years, not two, three or five before the drought event? It would be good to show the trajectory of BAI for each trials as the paper (Depardieu et al., 2020) et al. 2021) did in New phytologist.”

In the paper where the index was first described (Lloret et al. 2011), they chose a five-year period to characterize pre- and post-drought growth, in their words: “to avoid any overlaps with other low-growth periods”. In our study sites, 2010 was a relatively dry year with low overall growth so we decided to use the period 2011-2014 to avoid this low-growth year. The trajectory of BAI for each test site is shown in a new Figure in the Supporting information section (see S1 Fig).

“L202. It would be better if the distribution information could be shown in the supplementary file.”

The Density distribution for each studied trait was now included as Supporting information, see new S2 Fig. Please, note that now the original S1 and S2 Figs are, for this revised version of our manuscript, the S4 and S5 Figs, respectively.

“L204, please remove Tan et al. 2016 in here”

Done.

“L213, where is median?”

The word “median” was removed from the caption of Table 2.

“L225. a maximum missing data proportion for individual or locus? a minor allele count of one? Why?”

The maximum missing ratio was calculated with respect to genetic loci. Individuals with extensive missing data would indicate quality issues associated with DNA and the extraction protocol, which was carefully quality controlled during the extraction. 

The G-matrix calculation used in this research is based on counting the changes in allelic states of two alleles (i.e., 11 00 is one unit greater than 10 00; 0 and 1 are the alleles). Although biologically attainable, genetic loci with more than one minor allele can introduce noise in such analyses, especially when the mutational distance between the alleles is unknown. For example, for a locus with three alleles (0, 1, and 2), the change of 01 11 is not the same as that of 01 12, when the mutational effect is not linear. Therefore, for robustness, we limited the minor allele count to one. 

“L224 and 226 the description was not sound logistic.

For example, after maximum site depth <=70, a minor allele count of for each locus, then it would be a maximum missing data proportion of 30%?”

The site read depth, indicative of how much sequencing information was available as read counts that support the genotypic information per SNP site, is different from the missing data ratio. In this research, in order to ensure genotypic data integrity, the control of read depth was applied for two reasons: 1) to ensure the quality of SNP calling, and 2) to eliminate homologs elsewhere in the genome from being collapsed into the same site. We removed the SNP data that showed more than 70 sequencing reads aligned to the site to avoid false-positive calls on heterozygotes from homologs. 

For a population of 100 individuals, a SNP site with 30% missing data would mean that 30 individuals failed to generate genotypic information for the SNP locus.

We therefore changed the "site depth" to "site read depth" to avoid further confusion. 

“L236 if the parental trees are from 99 to 204, will this reduce the accuracy of those additional parents as it may only have one progeny for each parent?”

The number of genotyped trees per mother had a range of 1-20, and per father 1-9. However, while 80 mothers have a number of genotyped trees greater than 9, 81 (out of 100) fathers have only 1-2 genotyped trees. For sure a lower breeding value accuracy will be obtained for parents with only a few (1-2) progenies. However, the breeding values and accuracy of the parents can only be estimated using pedigree information given that the parents were not genotyped. For example, using the pedigree-based multiple-site model and for the DBH trait, the average theoretical accuracy of a father is 0.47 while for mothers it is 0.73. For this trait, the following plot shows the incremental increase in the average accuracy of breeding values for parents with an incremental increase in the number of offspring per parent.

“L247: sigma^2 is the additive genetic variance”

Yes, sigma^2 is the additive genetic variance estimated by pedigree- or genomic-based relationship matrices. The word “additive” was added.

“L248 σ^2*I?”

The identity matrix was added.

“L258: [′ | ⋯ | ′] is a matrix?”

No, this is vector notation. For example, the vector of observations [■(y_i@⋮@y_j )] of order n × 1 can be written as [y_(i )^' |⋯| y_j^' ].

“L271: Based on your genotypic data is from GBS genotyping platform, have you considered using other methods, such as Godds’ method or others to estimate pair-wise relationships (Dodds et al., 2015)? In this study, what imputation method did you use before you estimate the G matrix? As we know, GBS is quite sensitive for estimating relationships. Have you considered this?”

We consider GBS because it is a genotyping technology that shows consistency and robustness for spruce. We have proven GBS's effectiveness in interior spruce, white spruce, pine and other conifer species. 

Dodds et al. (2015) evaluated five relatedness measurements and concluded: “unbiased estimates of relatedness can be obtained by using only GBS SNPs”. Further, the KGD (GBS with depth adjustment) approach recommended in Dodds et al. (2015), calculated the G-matrix using only those SNPs which are scored in both of the corresponding individuals; and, when SNP genotypic values are not scored, missing values are replicated with zeros by assuming scored genotypes are a random sample across the genome. In this study, we encode SNP genotypic values as -1, 0, and 1. With the mean imputation conducted in this study, in theory, most missing values will be replaced by zeros. As a result, we have dealt with the missing data issue similarly, and believe the genomic relationship estimates in our study are robust.

In addition, and as we stated before, given the high number of SNPs used (> 460K) we feel that there should be subtle differences among algorithms. 

The imputation method used in this study is now clearly stated at the end of the “Genotyping by sequencing” Material and Methods sub-section.

“L356 the number of individuals selected in each trial is important.”

The number of trees selected for this study in each test site was added to Table 1

“L419: red and dark blue?”

Fig 4 show just two very low negative correlation values (-0.01 and -0.02). Therefore, the dot of cream color of these two negative values is almost negligible. Therefore, the legend of this figure now reads: “The estimated genetic correlations are shown in each cell below the diagonal, and the light to dark blue color of each individual cell above the diagonal reflects the strength of the genetic correlation.”.

“L466: Did you test your assumption for such higher heritability estimates for tree height and DBH? Based on the paper (Rweyoneze 2016), all families were measured at least in two trials in many years.”

We did not test the hypothesis that at age 30 HT generally produce higher heritability estimates than DBH. However, a preliminary study in this white spruce population showed that at age 30, DBH was under moderate to strong competition at the genetic level in the three test sites (direct and competition additive correlations from an additive competition model equal to -0.49 -0.79 and -0.85 for CALL, CARS and REDE, respectively). Previous work (for example see Hernández et al. 2019, For. Sci. 65: 570-580; Belaber et al. 2021, Ann. For. Sci. 78:2) showed that DBH is more sensitive to competition than HT. These studies showed that for a trait that revealed competition the competition model gave a better fit than simpler models, and the standard individual models reduce the additive variance and increase the error variance producing lower estimated heritability values as compared to the competition model.

The white spruce dataset used in this study have available HT measured at ages 8, 10, 15, 16, 21, 24, and 30, and DBH measured at ages 21, 24, and 30. A preliminary pedigree-based standard analysis for the common ages for both trait (21, 24 and 30), showed that the average across-sites heritability estimates for HT increased from age 24 to 30 for the three test sites (0.70 vs. 0.82 for age 24 and 30, respectively), while heritability estimates for DBH decreased (average across sites 0.50 vs 0.46 for age 24 and 30, respectively). These results showed that for a trait under strong competition (DBH), the standard model produce lower heritability, and demonstrated that HT have higher heritability estimates than DBH, and that these differences increased with the years.

“L467-468. I think you need to discuss more for such high heritability, not comparing the heritability between height and DBH. Only a small difference, I don’t think it is abnormal or need to discuss between height and DBH.”

We agree with the Reviewer comment that these heritability estimates differences, between HT and DBH, not abnormal and need no further discussion. Therefore, we removed this sentence from the original version of our manuscript. Additionally, and based on a comment of Reviewer #2 (see below), this paragraph (Lines 463-471 of the original MS) was simplified.

“L472-483 WD is lower than height and DBH? It is interesting for me.”

These estimates are not directly comparable as each trait is controlled by different sets of genes.

“What about other traits used in (Depardieu et al., 2020)? E.g. growth resilience, growth recovery, growth relative resilence?”

The calculation of these indices requires at least three years both before and after the drought event to calculate the average pre- and post-drought growth. The main drought events in the three sites during the study period occurred in 2002 and 2015. The first episode was too early to get accurate ring width data and the second episode occurred just before the samples were taken (although samples were taken in 2017, data only goes until 2016), so we were not able to calculate any of the other indices. 

“L513: 2008)”

To clarify this sentence one “(” was removed and other “)” was added.

“L536: Fig 3 and Table S1?”

The PLOS One format for citation of Tables in Supporting Information is “S1 Table” and no “Table S1”.

“L597: the results from Hannrup and Lenz et al. should be introduced together.”

Done.

“L672: delete Cappa et al. 2012”

Done.

“L697: what about Picea abies (Chen et al. 2014)?”

The results of this citation were added.

“L713: what are microenvironmental factors?”

These are factors that act within the trial level, i.e., spatial variation that acts at a small-scale such as soil depth, fertility, stoniness, humidity, etc.

“If the site was not under drought stress, it may be difficult to accurately estimate the value and genetic parameters.”

We agree with this comment that to accurately estimate these drought indices and their genetic parameters, we need to have a drought event with a strong effect on growth. Therefore, we have selected the drought event that occurred in 2015 that had a severe drought, despite the fact that this year (2015), and as we mentioned before, prevents us from calculating additional robust resilience indices. Drought stress of different degrees of intensity will produce different reactions in the trees, which is something that we already discussed in the original version of our manuscript (Lines 490-501).

 

Responses to Reviewer #2

“This manuscript shows quantitative genetic parameters in a subsample of a White spruce breeding population in the province of Alberta (Canada), estimated using both pedigree and genomic based relationships. The traits addressed here were of productive (growth and wood characteristics) but also of adaptive meaning (drought and pest resistance) in order to provide information for decision making within the breeding program for more productive and resilient spruce forests in the region.

This is a relevant manuscript for several reasons. Firstly, part of the novelty is because the genomic implementation in an operational tree breeding program. Although is not the first report about genetic estimations upon genomic predictions (see for example Ukrainetz & Mansfield 2020, Tree Genet & Genomes), it fits within the first stages of application of this new methodology in forest species such that new results and comparisons are much needed. Second, the array of traits is pretty much relevant and costly to get including x-ray wood density estimations, isotopic discrimination in timber tissues, dendrochronological parameters and chemical defenses. Thus, this work builds a unique dataset within the forest breeding programs field in order to address questions related to both breeding and ecological and evolutionary matters. Third, three environments have been tested, such that plasticity effects and genotype by environment interactions patterns can be disentangled. Finally, methodology and specifically the statistical analyses are sound and robust, given that from my perspective authors have deployed most modern and robust methodologies for quantitative genetics analyses in forest species, including the always-welcomed spatial corrections and proper linear mixed models.

Overall, the manuscript is well written and language is clear and stylish, it takes advantage of a relevant experimental design, implements an accurate methodology, introduces the topic properly and highlights the importance of addressing adaptive traits in forest breeding programs under current global change, shows materials and methods properly and take advantage of proper and updated bibliography. The manuscript is relevant for forest breeding science and its methodology is convincing. As consequence, I believe that the manuscript may be suitable for publication in ‘PlosONE’.

However, I am missing a more structured message, with a main message in front together with some other secondary messages. I have the feeling that the manuscript misses the opportunity to build a more meaningful message of interest for breeders and for even ecologists and evolutionary biologists interested in forest trees. Following are the symptoms I found within the manuscript which shows the lack of a particular and robust message:

“1. The title is ambiguous, and although it clearly shows what has been done it does not show what has been obtained.”

We considered the Reviewer’s comment about the title and modified accordingly. Now reads: “Integrating genomic information and productivity and climate-adaptability traits into a regional white spruce breeding program”.

“2. At the end of the introduction, no hypotheses are shown but authors point to the following goal: “L116-118. We studied 15 growth, wood quality, drought resilience, and defense and drought stress chemical traits (monoterpenes), and estimated their quantitative genetic parameters (including heritability and genetic correlations) within and across-sites”. Here, I do not agree that estimating genetic parameters per se is a goal for a scientific paper. Furthermore, in the same section author’s state: “L119-121. The results of this study would provide critical information for the identification and selection of genetic material for the production of productive, healthy, and resilient white spruce future forests”. Hence, if the critical information has been produced, why is not discussed in the current manuscript in terms of future strategies for the breeding program?”

We agree with the Reviewer and now the implications for the white spruce breeding program of the results obtained from this study are discussed in the new Discussion sub-section call “Implications for white spruce breeding”. The objectives of this manuscript were re-written at the end of the Introduction section to put more emphasis on the implications for the white spruce breeding program of the results obtained from this study.

“3. Although a relevant effort has been done in order to justify results and to put them in context in terms of current bibliography, large parts of the discussion are mainly comparisons of estimated genetic parameters with the ones obtained in other breeding programs and even for other species (e.g. L.545-560; L586-592; L.594-597). I believe that it is relevant to highlight that genetic parameters, as for instance heritability, are context dependent, as they depend on specific populations and specific traits in a specific time or environmental context. Thus, heritability comparisons with other populations or even other species and under different environmental conditions, although can be relatively useful to put in context some results, they have a limited relevance in order to provide meaning to a discussion.”

Several paragraphs of the “Trait genetic control” and “Relationship among traits” Discussion sub-sections were simplified or reduced. The paragraph in Lines 593-597 of the original version of our manuscript was removed. 

We are agree with the Reviewer that is important highlight that genetic parameters are context dependent. In this sense, we have add the following sentences at the top of the “Trait genetic control” Discussion sub-section: “Genetic parameters and function of them, as heritability, play an important role in the selection of parents in a breeding program. However, its values are context dependent, as they depend on the relative contributions of genetic and environmental variations in a specific population, and vary among traits and ages [50]”.

“Hence, given that the authors are not clearly showing a main message, the manuscript looks like unfocused, novelty is not clearly stated (too many comparisons with former works such that it looks like that everything has been done before) and may end in reduced interest for a potential audience. Furthermore, the results section is quite difficult to follow given the big list of genetic parameters estimated in 3 different sites for 15 different traits, including significance and standard errors.”

The three sub-sections of the Result were simplified, i.e., several sentences from theses sub-sections of the original version of our manuscript were removed.

My recommendation to the Editor is to ask for a major revision such that the manuscript can be rewritten to become more centered in specific messages, to attain higher scientific quality and become more helpful for potential readers.

Following I am attaching some thoughts and ideas with the aim to inspire authors and to let them know where is the gap that in my opinion exists:

“1. Discussion is too long and does not help to center the message. For instance, some paragraphs as L.463-471 are not novel results at all in forest breeding, thus I believe that it should be removed in order to make proper room for the relevant messages. Instead, paragraphs as L.598-614 are a good example of a more message-centered discussion that may be a reference to rewrite discussion.”

As we answered previously (see answer of previous item 3), several paragraphs in the “Trait genetic control” and “Relationship among traits” Discussion sub-sections were simplified, reduced, o eliminated in the current version of our manuscript.

“2. A specific goal should be highlighted from the very beginning, whether it is to show a discussion about future breeding strategies within the breeding program upon current results, or to center the discussion in evolutionary and physiological trade-offs among adaptive and growth traits or even a combination of both perspectives. From my point of view, the traits measured here have a pretty big room for physiological and evolutionary discussion what is enhanced by the selection of a wide random population with half-sibs structure tested in 3 sites.”

As we answered previously, a new Discussion sub-section call “Implications for white spruce breeding” was added. In this sub-section we focus the discussion on the implications for the white spruce breeding program of the results obtained from this study.

“3. Other potential focus of discussion for the manuscript in the comparison between pedigree and genomic based relationships estimations, given that I have the feeling that it has not been properly discussed even if the topic is shown in the title.”

We agree with the comment of the Reviewer and, therefore, a sub-section titled “Pedigree- and genomic-based relationship estimations” has now been added at the beginning of the Result section. Please note that a new Table 3 and a new S3 Fig were also added. Additionally, a new paragraph, discussing differences in estimates of heritability from pedigree- and genomic-based relationship matrices, was added at the end of the “Trait genetic control” Discussion sub-section. Differences between genetic correlation estimates resulting from using pedigree or genomic relationship information were already discussed in the previous version of our manuscript.

Finally, following I point specific mistakes, advices or concerns:

“L.63-65. References are missing.”

As we answered previously to Reviewer #1, three references were added.

L.88-90. I believe this statement is too risky. Authors should be more explicit in terms of what actually has not been studied before.

The sentence was removed.

L.107-109. Again, I believe this statement is too risky. Authors should be more explicit in terms of what actually has not been reported before.

The sentence was removed.

L.117. Here it should be shown the main goals, and also some hypothesis if needed.

As we answer previously, the objectives of this manuscript were re-written at the end of the Introduction section to put more emphasis on the implications for the white spruce breeding program of the results obtained from this study.

“L.129. “Testing population” does mean the original breeding population that has been subsampled? State it clear.”

This has been rewritten to state that the entire population tested in the progeny trials consisted of 150 families. The families (80) we studied in depth, were selected from these 150 original families.

“L.192. If I am not wrong it should be -40ºC, right?”

Correct, the needle samples were kept at -40C. This has been corrected.

“L.242-243. Which is the reason for accounting for the provenance effect as a fixed effect in the model? Based on the goals that seem to be sought, should not be better to concentrate the whole genetic variation in the family effect for breeding purposes?”

The fixed effects genetic groups formed according to provenances were fitted to minimize the bias in quantitative genetic parameter estimates derived from the potential genetic heterogeneity among trees with unknown parents in the pedigree or some underlying provenance structure not captured by the genomic information.

“L.245. Please list the whole list of random effects for clarity.”

All the random effects were included clearly as a term in the model [1], the replication design effect (d) and the random genetic effects (a).

“L.263. A further explanation about ‘A’ is needed.”

Done.

“L.267. A further explanation about ‘I’ is needed.”

Done.

“L.452-455. My impression is that in forest breeding programs over the last 20 years the most relevant target traits after productivity are pest and disease resistance. Although I am agree that in terms of drought resistance we still are in very preliminary stages, I believe authors should rewrite this sentence or make a further rationale justifying the statement.”

We agree with the comment from the Reviewer about that growth is a priority trait and after pest and disease resistance, drought resistance is a relatively less studied trait in conifer breeding. In a recent LIA FORESTIA webinar (https://www6.inrae.fr/forestia/SEMINARS) titled “Eulogy of fast growth… up to what costs? A preliminary recognition before a study of trade-offs of growth heterosis in hybrid larch”), Dr. Luc E. Pâques (INRAE- UMR Biofora Orléans) showed a slide that we can see that over 6,961 conifers breeding studies found on the web, 3,338 of the studies focused on growth traits, 842 on wood property traits, 302 on disease and pest resistance, and only 141 on drought traits (see Pâques´ slide below). In summary, we modified this sentence as suggested by Reviewer #2, that now reads: “While the need to identify adaptability-related trait genotypes grows, less effort has been directed towards the selection of pest and disease resistant trees, and even less for the selection of drought resistant/resilient individuals.”.

“L.459-461. If “valuable information is expected to be provided to bred and assist in selection”, why authors do not discuss this information in terms of breeding strategies?”

As we answered previously, a new Discussion sub-section call “Implications for white spruce breeding” was added. In this sub-section we focus the discussion in terms of breeding strategies.

“L.459. “Breeds” should be “breed””

Done.

“L.526-527. I am not totally sure about this statement. Moreira, Sampedro, Zas and collaborators for instance have amply published about this question.”

This sentence refers to studies that have focused on estimation of heritabilities of secondary compounds in forest tree. We see on the web that Moreira, Sampedro, Zas and collaborators have published several studies that showed the relationship between chemical secondary compounds and herbivore attack or drought stress in conifers, but not including heritability estimations of these compounds. For clarity, this sentence now reads: “In spite of the importance of these compounds in relation to adaptability-related traits, few studies have focused on the genetic control (i.e., heritability estimates) of secondary compounds in forest trees.”.

“L.534. “breeders” instead of “breeds”.”

Done.

“L.570. Delete “his studied”?”

Deleted.

“L.622. “..in dry years δ13C the correlation increased..” something is wrong here.”

We regret we have not expressed ourselves clearly enough with this sentence which now reads: “For wet years, WD and δ13C was negatively correlated and, in dry years δ13C increased with increasing wood density (i.e., positive correlation)”.

“L.628. “resistant” instead of “resistance”.”

We are referring to our drought index which is called Resistance not Resistant. Therefore, we prefer to keep this sentence as originally written, i.e., “Resistance and Sensitivity drought indices were marginally negatively or positively correlated with …”.

“L.665-666. Why authors did not applied same methodology they advise?”

The application of single-step GBLUP models is certainly a worthwhile endeavour to apply to all the studied traits. However, the application of this approach in multiple-trait individual-tree mixed models as the fitted in this study are prohibitively expensive in both computation time and memory requirements.

“L.672. Citation “Cappa et al. 2012” is repeated twice.”

Deleted.

“L.683. Delete “in” in “lower in annual precipitation…”?”

Done.

“L.720-722. Baltunis et al (2008) showed low genetic correlation and strong G×E as consequence. Hence, this citation here together with other studies which found no significant G×E like Guy and Holowachuk (2001) does not match. It should be together Cregg et al. (2000) who observed strong G×E.”

The Reviewer is correct. It was a mistake in our previous version of the manuscript which has now been corrected.

“Table 2. Min. and Max. values for ‘HT are in the wrong units.”

The Reviewer is correct. It was a mistake in our previous version of the manuscript and has been corrected.

“Fig. 2. This is not the proper Figure. Actually, Fig.2 in main text is exactly the same than S1 Fig. which show genetic correlations and not heritabilities as it should be.”

The Reviewer is correct. It was a mistake in our previous version of the manuscript and has been corrected.

---

## [Decision Letter · Decision Letter 1]

14 Feb 2022

Integrating genomic information and productivity and climate-adaptability traits into a regional white spruce breeding program

PONE-D-21-25537R1

Dear Dr. Cappa,

We’re pleased to inform you that your manuscript has been judged scientifically suitable for publication and will be formally accepted for publication once it meets all outstanding technical requirements.

Kind regards,

Ricardo Alia, PhD

Academic Editor

PLOS ONE

Additional Editor Comments (optional):

Dear author, thanks for addressing all the comments made by the referees. I think that now the paper can be accepted for publication.

Reviewers' comments:

Reviewer's Responses to Questions

**Comments to the Author**

1. If the authors have adequately addressed your comments raised in a previous round of review and you feel that this manuscript is now acceptable for publication, you may indicate that here to bypass the “Comments to the Author” section, enter your conflict of interest statement in the “Confidential to Editor” section, and submit your "Accept" recommendation.

Reviewer #2: All comments have been addressed

2. Is the manuscript technically sound, and do the data support the conclusions?

Reviewer #2: Yes

3. Has the statistical analysis been performed appropriately and rigorously? 

Reviewer #2: Yes

4. Have the authors made all data underlying the findings in their manuscript fully available?

Reviewer #2: Yes

5. Is the manuscript presented in an intelligible fashion and written in standard English?

Reviewer #2: Yes

6. Review Comments to the Author

Reviewer #2: I acknowledge authors for addressing the whole questions and concerns raised in the former review. Authors have thoroughly answer every question raised and have improved the manuscript following reviewers guidelines. Former version of the manuscript already met several publication requirements for Plos One: specific questions are addressed, novelty, relevant experimental design and accurate methodology. Therefore, main outputs from the manuscript are relevant for forest breeding science and are convincing. Furthermore, authors have improved the manuscript by rewriting several sections such that new Discussion section makes this work more meaningful and relevant.

Hence, from my point of view, this manuscript is ready for publication in Plos One.

Following I show some specific mistakes I still found in the text:

L.248. Last bracket should be deleted

L.693. ‘suggesting that trees more resistance to a drought’ is not right, it should be ‘resistant’.

L.826. ‘determines’ instead of ‘determine’.

7. PLOS authors have the option to publish the peer review history of their article (what does this mean?). If published, this will include your full peer review and any attached files.

Reviewer #2: **Yes: **Dr. Raul de la Mata

---

## [Editor Report · Acceptance letter]

9 Mar 2022

PONE-D-21-25537R1 

Integrating genomic information and productivity and climate-adaptability traits into a regional white spruce breeding program 

Dear Dr. Cappa:

I'm pleased to inform you that your manuscript has been deemed suitable for publication in PLOS ONE. Congratulations! Your manuscript is now with our production department. 

Kind regards, 

on behalf of

Dr. Ricardo Alia 

Academic Editor

PLOS ONE